# Recent Advances and Mechanisms of Phage-Based Therapies in Cancer Treatment

**DOI:** 10.3390/ijms25189938

**Published:** 2024-09-14

**Authors:** Vivian Y. Ooi, Ting-Yu Yeh

**Affiliations:** Agricultural Biotechnology Laboratory, Auxergen Inc., Riti Rossi Colwell Center, 701 E Pratt Street, Baltimore, MD 21202, USA

**Keywords:** phage, peptide display, cancer therapy, gene therapy, immunity

## Abstract

The increasing interest in bacteriophage technology has prompted its novel applications to treat different medical conditions, most interestingly cancer. Due to their high specificity, manipulability, nontoxicity, and nanosize nature, phages are promising carriers in targeted therapy and cancer immunotherapy. This approach is particularly timely, as current challenges in cancer research include damage to healthy cells, inefficiency in targeting, obstruction by biological barriers, and drug resistance. Some cancers are being kept at the forefront of phage research, such as colorectal cancer and HCC, while others like lymphoma, cervical cancer, and myeloma have not been retouched in a decade. Common mechanisms are immunogenic antigen display on phage coats and the use of phage as transporters to carry drugs, genes, and other molecules. To date, popular phage treatments being tested are gene therapy and phage-based vaccines using M13 and λ phage, with some vaccines having advanced to human clinical trials. The results from most of these studies have been promising, but limitations in phage-based therapies such as reticuloendothelial system clearance or diffusion inefficiency must be addressed. Before phage-based therapies for cancer can be successfully used in oncology practice, more in-depth research and support from local governments are required.

## 1. Introduction

Despite the many advancements achieved in clinical medicine, there are still diseases which continue to plague modern scientists. One of the most challenging of these diseases is cancer, with as many as one in five people in the world developing cancer in their lifetime [1]. Cancer is a disease characterized by the uncontrolled growth of abnormal cells and their spread (metastasis) to other parts of the body [2]. The complexity of cancer, particularly the molecular and genetic variability across its subtypes, makes it challenging to treat and ultimately cure. While there are a range of current treatments such as surgery, chemotherapy, radiation therapy, personalized targeted therapy based on next generation sequencing findings, and immunotherapy, scientists are searching for new therapies that can overcome drawbacks including poor accessibility, side effects, lack of effectiveness, and high cost. An example of a burgeoning treatment option is phage-based therapy. This relatively novel application of bacteriophages is a promising avenue of research and is being further explored for its great potential to become a reliable treatment option for cancer.

## 2. What Are Bacteriophages?

Bacteriophages, one of the most abundant biological entities, are a diverse set of viruses that infect bacteria. The International Committee for Taxonomy of Viruses (ICTV) has classified phages into twelve families, including *Straboviridae*, *Autographiviridae*, and *Drexelvirida* [3,4]. The classification by the ICTV for bacteriophages relevant to clinical cancer treatments is summarized in Table 1. Phages were hypothesized by Frederick Twort in 1915 and later discovered by Félix d’Hérelle in 1917 [5]. Very little was known about phages when they were discovered, and it was only until 1940 when phages were first visualized through electron micrographs in Germany [6]. The early years of phage research focused on only a few types of phages, most notably the phages that infect the bacteria *Escherichia coli*. As familiarity with these viruses grew, bacteriophages began to be utilized in experiments, some of which were important investigations that enriched our understanding of integral molecular biology concepts such as genetic material, the 3-nucleotide code, and restriction enzymes [5]. 

After years of research, the fundamental properties and mechanics of phages are now better understood. These powerful nanoparticles range in size from 40 to 200 nm (although filamentous phages are a few micrometers) [7]. Since structure is not compromised by their small size, phages are suitable for application in nanomedicine as different structural components can be manipulated through bioengineering. Genetic material, either single or double-stranded DNA or RNA, is enclosed in a protective capsid with exterior coat proteins. Basic forms are an icosahedral head (with or without a tail) or the filamentous form (which is not confined by a cap-like structure) [8]. Phages can have a narrow or broad host range (the diversity of organisms that the phage can infect). For example, JHP phage can infect *P. aeruginosa*, *E. coli*, *S. enterica*, *Campylobacter jejuni*, *Acinetobacter baumanii* and *Proteus mirabilis*, whereas M13 phage can only infect *E. coli* [9]. 

Infection is initiated when the phage interacts with receptors in the cell wall, capsular polysaccharide, pili, or flagella (varying between phage species). Phage absorption rate is governed by the frequency of collisions, which increases as the concentration of the virions and bacterial cells increases. The phage then ejects its genetic material into the cell [10].

Phages can be divided into two types based on mode of replication: virulent and temperate. Virulent phages undergo the lytic cycle in which the phage attaches to the host, the phage genome is inserted into the cytoplasm, and multiple phages are reproduced using host cell machinery. Nascent particles are released through lysis and can infect other host cells. Temperate phages undergo the lysogenic cycle, in which phage DNA is integrated into host DNA as a prophage and is passed on to daughter cells as the host divides. Temperate phages can switch to the lytic cycle under certain environmental conditions, such as UV irradiation or whether there are sufficient co-infecting phages generated [11,12,13]. In both cases, large amounts of DNA are encapsidated in nascent particles and the genetic material is stabilized to prevent degradation [14].

Although phages infect bacteria, they can also interact with eukaryotic cells. The earliest report of phage interaction with tumor tissue was done in 1940 by Bloch et al. who observed that phages could accumulate in Erhlich carcinomas and inhibit tumor growth [15]. Research has led to a general understanding of phage-eukaryotic cell interactions which thus prompted the proposal of phage treatment for human cancer. Phages can enter eukaryotic cells through processes like receptor-mediated endocytosis, transcytosis, clathrin-mediated endocytosis, macropinocytosis, or caveolae-mediated endocytosis, but this depends on the type of eukaryotic cell and phage [16]. While phages do not infect and replicate within eukaryotic cells [17], they may still interact with intracellular proteins such as Toll-like receptors and cytosolic proteins [16].

## 3. Phage-Based Therapy

Phage therapy as a form of treatment for humans was initially controversial due to the lack of understanding regarding phage structure and mechanisms [18]. Early phage therapy trials had poor documentation and were met with varying degrees of success. In fact, Western interest in bacteriophage therapy had dissipated by 1940 because most doctors at the time were “solo general practitioners” who did not have access to the resources needed for the implementation of phage therapy [19]. Furthermore, the period was dominated by chemical compound-based treatment methods like sulfas and antibiotics, which brought along substantial profits for pharmaceutical companies. Despite Western disinterest, countries in the former Soviet Union (most notably Georgia and the G. Eliava Institute of Bacteriophages, Microbiology, and Virology) continued pursuing research into phages and phage therapy [3]. In the past few decades, though, a global interest in bacteriophage therapy and its applications has been restored owing to issues threatening the sustainability of some therapies (like antibiotic resistance).

In the context of cancer therapy, phage-based therapies have theoretical advantages over conventional types of oncology treatment or combinations of them (surgery, chemotherapy, radiation therapy), which could alleviate side effects and improve patient outcomes. The main advantage is the high specificity of phages for their target host [20]. Current chemotherapy has lower specificity as it destroys all rapidly proliferating cells, regardless of whether they are tumorigenic or healthy [21,22]. Enhanced selectivity can reduce the side effects that are commonly associated with chemotherapeutic agents (such as nausea, vomiting, hair loss, decreased appetite, and bone marrow suppression) [21] by attacking only those cells that display a certain marker and leaving healthy cells alone. Alternatively, a more holistic approach of “training” the immune system to recognize specific markers on cancerous cells could serve as natural long-term protection against possible metastasis or recurrence. Another advantage is reduced toxicity: phages are mostly made of nucleic acids and proteins, making them inherently non-toxic. This is a major advantage over chemotherapy and radiation therapy, which expose the body to toxic chemicals and rays. Phages are also highly versatile and can carry gene-editing tools alongside the antigen to treat cells at the genetic level, a feature not found in conventional therapies which mostly destroy cancer cells [20]. The “single-hit kinetics” of phages, meaning only one phage is needed to target one cell, allows for fewer units of phages per treatment compared to their chemical counterparts. Among other advantages, phage particles can also remain stable in a broad pH range, have relatively low cost of production, stay in storage for long periods of time, and have smaller environmental impact when they are discarded (compared to chemicals) [20,23].

Due to phages being viruses, employing phage-based therapies in humans can raise questions of safety. However, phages are unable to replicate in eukaryotic cells which quells concerns of viral infection within the mammalian host [17]. Additionally, data from a human trial show that human microbiota remains largely unaffected from oral administration of phages, addressing concerns that introducing phages into the human body could kill natural gut bacteria [24]. In fact, phages are naturally found in the human body. Phages (mostly temperate) that were found in the intestines belong to families of double-stranded DNA like *Siphoviridae*, *Myoviridae* and *Podoviridae*, but also to those of single-stranded DNA like *Microviridae* and *Inoviridae* [25,26]. Studies of phage mechanics in the human microenvironment show that interaction with the human body is mostly unrestricted, with transcytosis in human tissue and crossing of the cell membrane possible, preventing disruption of basic functions [27,28].

Production of these antigen-specific phages is straightforward thanks to phage display, a technique which has revolutionized the application of phages in biotechnologies. In phage display, the genetic material which codes for a target polypeptide is fused with genes that express coat proteins, thus allowing the polypeptide to be expressed on the surface of the phage [29,30,31]. An example polypeptide is the HER2 antigen, which can induce anti-HER2 antibodies in breast cancer treatment. Using this technique, scientists can create a large collection of phages (called a library) displaying different types of proteins. Due to the manipulability of phage coat protein expression, phage display libraries have been used to isolate target proteins, study protein-protein interactions, and identify bacterial strains [31]. Phage display can overcome lack of specificity in cancer treatment by leveraging antigen-receptor interaction and serving as a versatile vehicle for targeted delivery.

## 4. Main Mechanisms of Phage-Based Therapies

The versatility of phages has allowed scientists to explore different ways to exploit their characteristics for use in cancer treatment. To summarize, the two main mechanisms of phage-based cancer therapies are to use phages to elicit a form of immune response or to use phages as transporters of some molecules (Figure 1).

The first mechanism is for the phage to elicit an immune response to destroy target cancer cells (Figure 1A). Phage display allows mass display of tumor-specific antigens with potential for vaccine development. Once in the human body system, these bacteriophages are phagocytosed and decomposed by antigen-presenting cells (APCs) [32]. The antigen is fragmented into smaller peptides before being presented through two general major histocompatibility complex (MHC) paths: the MHC class I or the MHC class II pathway [33]. APCs can present the antigen by MHC class II molecules which interact with CD4+ helper T cells. The helper T cells pass along the antigen information to B cells, activating the B cells and encouraging production of antibodies that can bind to tumor cells with the same antigen on their cell surface (Figure 1(A.1)). This cell-antibody interaction can activate the complement system, prompt destruction by natural killer cells, or encourage macrophagic phagocytosis [34]. Alternatively, APCs can present the antigen by MHC class I molecules which interact with CD8+ cytotoxic T cells. The cytotoxic T cells acknowledge the antigen and use this information to attack tumor cells displaying the same type of antigen through secretion of proteins such as granzyme, perforin, cathepsin C and granulysin to cause cell death [35] (Figure 1(A.2)).

Phages can also be used as transport vehicles for targeted delivery (Figure 1B). The goal of targeted delivery is to deliver treatment directly to target cells [36]. Since phages can be engineered to display certain peptides which recognize specific receptors, such as the RDG4C ligand which binds to α_v_β_3_ and α_v_β_5_ integrins on cancer cells, they serve as good carrier candidates. One molecule that phages can deliver is photosensitizers to target cells as part of photodynamic therapy [37]. Upon activation by light at a certain wavelength, these photosensitizers react with oxygen to form reactive oxygen species which can kill cancer cells (Figure 1(B.1)). Cytokines can also be displayed on phages to trigger immune responses including stimulation of macrophage inflammatory response [38] (Figure 1(B.2)).

But perhaps the most promising form of targeted delivery is gene therapy, which is the transfer of engineered genetic material into cells to alter cellular responses and treat diseases [39] (Figure 1(B.3)). Limitations of gene therapy include targeting inefficiency, vector instability, and infection risks from viral vectors [40]; however, they can be overcome by using phages because of higher specificity through phage display, sound structure, and inability to infect mammalian cells. Initially, eukaryotic viruses were regarded as a promising vehicle for transfer of genes into human cancer cells due to high transfection efficiency and stable expression of transferred genes, but several problems came to light such as their absorption by non-target tissues [7]. To reap benefits of both forms, the idea of using eukaryotic and prokaryotic viruses was combined. Most studies create a chimeric bacteriophage with a modified exterior peptide to target cancer-specific receptors as well as an encoded eukaryotic viral transgene to trigger a cellular response.

Cellular responses can have different pathways depending on the eukaryotic viral transgene delivered. Forms of genetic therapy commonly seen in cancer treatment include insertion of an apoptosis-inducing gene, replacement of a faulty gene, or prodrug activation therapy. A gene encoding for a particle that induces apoptosis can be transfected into target cells with the help of specific antigen-receptor interaction [41]. Once the gene is transcribed and translated, the resulting protein signals downstream pathways and causes cell death. One commonly delivered gene encodes the cytokine tumor necrosis factor-related apoptosis-inducing ligand (TRAIL). Secreted TRAIL binds to membrane receptors DR4 and DR5, which activate caspases or induces mitochondrial-dependent death [42] (Figure 1(B.3.a)). A faulty, mutated gene can also be replaced by inserting a functional copy: this can be seen with gene therapies targeting p53 where a gene encoding functional p53 is delivered to cancer cells with mutated p53 (which has lost its tumor suppressor functions) [43] (Figure 1(B.3.b)). In prodrug activation therapy, a gene encoding an enzyme and genetic material needed for its transcription is inserted into cells [44]. The enzyme can activate a prodrug into a cytotoxic agent through metabolization (Figure 1(B.3.c)).

## 5. Current Phage-Based Therapies across Various Cancer Types

To provide an overview of the updated advancements in phage therapy research on different cancers, the most recent studies published for each cancer type have been compiled and summarized. Current experimental phage-based therapies across different cancer types are summarized in Table 2.

### 5.1. Breast Cancer

In breast cancer, phage-based therapy is being applied to target *HER2* oncogene, a membrane receptor tyrosine kinase that is overexpressed in cancer cells. HER2 activates various signaling pathways upon dimerization (i.e., MAPK/ERK and PI3K-Akt pathways) that lead to cell proliferation [45]. Although the advent of trastuzumab and other targeted therapy drugs has improved the prognosis and clinical outcome of HER2-positive breast cancer patients, drug resistance remains a challenging issue that necessitates alternative forms of treatment.

Development of phage-based therapy for breast cancer is showing significant progress, with the latest experiments testing phage-based vaccines on mice. Barati et al.’s study in 2018 [46] featured an engineered lambda bacteriophage to treat mice with the TUBO cell line model of breast cancer. Every two weeks for a total course of six weeks, the mice were immunized with λF7 phage that was engineered to display an AE37 peptide fused to the gpD coat protein. AE37, derived from human HER2, is an MHC class II hybrid molecule that stimulates CD4 and CD8 cells [47]. Compared to control groups (λF7 without *gpD::AE37* gene and TN buffer), the percentage of CD8+ T cells was significantly greater in the mice injected with λF7 (gpD::AE37), indicating the importance of the added peptide in eliciting an immune response. In both the prophylactic and therapeutic studies, mice injected with λF7 (gpD::AE37) showed greater median survival time, time to reach the end point, and delay in tumor growth compared to the controls [46].

Similarly, Wang et al.’s study [48] utilized a M13 bacteriophage-based vaccine to treat female mice with mammary adenocarcinomas (using BT-474 cell line). M13 phages were engineered so that HER2 immunogenic epitopes (the extracellular and transmembrane domains of HER2 or its splice variant Δ16HER2) were attached to phage coat pIII. Both HER2 and the variant were effective in both a prophylactic and therapeutic setting, delaying tumor onset and decreasing tumor growth rate. Within 15 weeks of age, all female mice treated with empty phages developed palpable tumors, while 75% of mice vaccinated with ECTM-phages and 40% of mice vaccinated with Δ16ECTM-phages were still tumor-free. Purified antibodies from both HER2 and the variant showed interference with the ERK/MAPK signaling pathway (which promotes cell cycle progression) and reactivation of the retinoblastoma tumor suppressor. There was also a reduction of cancerous cell viability in trastuzumab-resistant human breast cancer cells when treated with IgG of both epitopes, showing that the M13 vaccine can serve as an alternative to patients who are resistant to trastuzumab.

In 2008, Shadidi et al. [49] showed that oral immunization with T7 phage displaying the tumor antigen Hsp27 could lead to the induction of effective immune responses in 4T1 murine breast cancer cells which overexpress Hsp27. Mice were subcutaneously injected with 4T1 cells on day 30 after immunization with the T7 phage expressing the Hsp27. Mice orally immunized with T7 phage expressing the Hsp27 developed tumors that had a significantly smaller average weight versus the wild-type phage (1.5 ± 0.2 g versus 2.5 ± 0.5 g; *p* < 0.05). Additionally, mice immunized with the wild-type phage had a significantly higher number of lung metastases than the Hsp27 phage treatment (80 ± 15 vs. 40 ± 10, *p* < 0.05). These data provided the first evidence that a recombinant T7 phage stably expressing Hsp27 can inhibit the outgrowth of 4T1 breast cancer cells.

In Pouyanfard et al.’s 2012 study [50], multivalent T7 bacteriophage nanoparticles were developed that displayed an immunodominant H-2k^d^-restricted CTL epitope (9-mer peptide p66 TYVPANASL) derived from the rat HER2/neu. The authors tested the therapeutic potential of these chimeric T7 nanoparticles in 6- to 8-week-old female BALB/c mice. Only T7 phage nanoparticles carrying a single copy of p66 peptide were successfully cross-presented and elicited a specific T cell response against the displayed CTL epitope. Immunization with these nanoparticles induced CTL-mediated lysis of P815 target cells in vitro and increased IL-4 cytokine secretion. T7-p66-vaccinated mice efficiently rejected HER2-expressing TUBO Cells. Moreover, anti-tumor effects and dramatic tumor regression were significant in some mice vaccinated with p66 + FA, whereas all control mice developed large tumors by day eighty.

### 5.2. Colorectal Cancer

Colorectal cancer (CRC), the second leading cause of cancer death in the world, is usually treated with surgical resection, chemotherapy, and radiation. Metastasis to organs like the liver and the lungs continues to be a challenge for CRC [51]. In addition to routine screening for early detection through fecal occult blood examination, colonscopy, and sigmoidoscopy, researchers are looking at new approaches deviating from the standard treatment regimen.

Wang et al.’s study in 2021 [38] featured delivery of cytokine GM-CSF through expression on pVIII of M13 phage to increase the antigen expression of macrophages and dendritic cells. The variant allowed the display of multiple copies of a large protein so that there was a greater number of cytokine GM-CSF per phage. Experiments conducted in murine macrophage (cell line RAW264.7) revealed that GM-CSF presence ultimately led to phosphorylated STAT5, which plays a role in apoptosis and cell proliferation. The effect of GM-CSF displayed on phage was also investigated in a murine model (BALB/c mice inoculated with CT26 CRC cells), where therapy was administered four times every other day. GM-CSF showed significant reduction in tumor size compared to wild type phage. Combined radiation and phage therapy showed increased IFN-g expression in CD4+ T cells, CD8+ T cells, macrophages, and neutrophils compared to solely one or the other.

Turrini et al.’s study [52] used an engineered M13 phage displaying a tumor-specific nonamer peptide (CPIEDRPMC) to deliver the photosensitizer Rose Bengal (RB) to colorectal cells for treatment with photodynamic therapy. One challenge of photodynamic therapy is the low specificity of photosensitizers toward tumor cells which can cause side effects: by using phages to directly deliver RB to target cells, uptake of RB by healthy cells can be prevented. Upon light irradiation, HT29 cells showed a prominent decrease in cell viability and evidence of intracellular reactive oxygen species. When the bioconjugate was tested on the HCT116 cell line which was known to have low affinity to CPIEDRPMC, cells were not killed as efficiently, demonstrating that the presence of the targeting peptide significantly improved specificity. Experiments done on colorectal spheroids showed not only deep penetration of the nanovector into the spheroid core but also disruption of tumor structure.

### 5.3. Lung Adenocarcinoma

Lung adenocarcinoma is a common subtype of non-small cell lung cancer, arising due to tobacco smoke exposure, environmental factors, and genetic conditions [53]. Several tyrosine kinase inhibitors (TKIs) are currently available as treatment for EGFR mutation positive patients [54]. However, patients with non-small cell lung cancer due to a commonly mutated tumor suppressor gene, *p53*, may be more resistant to chemotherapy and radiation. To combat this, *p53* gene replacement therapy is being tested although an efficient vector is needed [55].

In Yang et al.’s study [56], an M13 bacteriophage was used as a vehicle to deliver a CRISPR-Cas9 transgene cassette (which was flanked by inverted terminal repeats of the adeno-associated virus 2) to lung adenocarcinoma cells. The bacteriophage was engineered to display the RGD4C ligand which can bind to overexpressed integrin α_v_β_3_ and α_v_β_5_ in tumor cells. This vector targeted A549 cells with 60–70% efficiency, as quantified by the number of cells expressing Cas9 protein. Most importantly, HEK293 cells with CRISPR-Cas9 expression had little to no p53 protein expression while those that lacked CRISPR-Cas9 expression showed continued p53 protein expression, demonstrating therapeutic value. This M13 phage construct is a hopeful solution to the gene delivery vector problem surrounding *p53* gene replacement therapy.

In Ren et al.’s study [57], a T4 phage nanoparticle expressing mouse Flt4 (mFlt4), which promotes tumor metastasis by stimulating solid tumor lymphangiogenesis, was constructed to evaluate the phage’s antitumor activity. The effect of the T4-mFlt4 vaccine was explored in mice injected with cells of Lewis lung carcinoma (LLC), a VEGFR-positive tumor. Mice with LLC-derived tumors showed extended survival when treated with the T4-mFlt4 vaccine compared to the control group. The treated group had a survival duration of 64 days, compared to 29 and 33 days in the control groups. By day 62, the survival rate in the T4-mFlt4 group was 25%, while the control phage group had a survival rate of 12.5% at 29 days. The vaccine did not demonstrate an ability to inhibit tumor growth, but it suppressed tumor metastasis (0.45 g ± 0.12 g versus 0.84 g ± 0.29 g in the control, *p* = 0.002) in mice. Histochemical examinations showed that the mean lymphatic microvessel counts were reduced in tumors after T4-mFlt4 treatment, but the average vascular microvessel counts in lungs were not significantly different between the groups.

Ren et al.’s 2011 study [58] demonstrated the protective immunity of the T4-mVEGFR2 vaccine against Lewis lung carcinoma (LLC) in mice. In vitro studies showed that immunoglobulin induced by the T4-mVEGFR2 inhibited VEGF-mediated endothelial cell proliferation and tube formation. The antitumor activity was further confirmed through adoptive transfer of the purified immunoglobulin. The antitumor activity and production of autoantibodies against mVEGFR2 were neutralized when CD4+ T lymphocytes were depleted.

Zuo et al.’s 2019 study [59] evaluated the anti-angiogenic effects of recombinant T4 phages expressing the extracellular domain of VEGFR2 (T4-VEGFR2). The T4-VEGFR2 phages specifically bound to VEGF and inhibited VEGF-induced phosphorylation of VEGFR2 and downstream kinases, such as extracellular signal-regulated kinase (ERK) and p38 mitogen-activated protein kinase (MAPK). In vitro experiments demonstrated that T4-VEGFR2 phages could inhibit VEGF-induced endothelial cell proliferation and migration. The administration of T4-VEGFR2 phages suppressed tumor growth, reduced microvascular density, and prolonged survival in murine models of LLC and colon carcinoma (CT26 cell line).

### 5.4. Hepatocellular Carcinoma

Hepatocellular carcinoma (HCC) is a hypervascular liver tumor and the sixth most common cancer worldwide [60]. Since chemotherapy does not improve survival rates, current treatments include tumor resection, liver transplant, or sorafenib [61]. Patients in later stages cannot benefit from surgery or transplant due to metastases and can only turn to nonsurgical approaches, such as transarterial embolization, transarterial chemo- embolization, and ablation therapy. Due to the less favorable outcome, HCC is one cancer type where phage-based therapy appears as new hope to patients [61].

Iwagami et al.’s study [62] investigated the effect of an engineered λ phage-based vaccine on developing immunity against HCC in mice. The BNL cell line overexpresses aspartate β-hydroxylase (ASPH), a protein which contributes to tumor cell proliferation by inhibiting apoptosis and delaying cell senescence. To target BNL cells, two forms of λ phage were engineered, one expressing a sequence derived from the N-terminal of ASPH (form λ1) and the other a sequence derived from the C-terminal of ASPH (form λ3). Compared to an empty phage control, prophylactic immunizations of both forms followed by a boost every 7–10 days inhibited tumor growth. In the in vitro experiments, activation of ASPH-specific CD4+ and CD8+ T cells and higher Th1 and Th2 cytokine levels (compared to the control) were observed, suggesting successful stimulation of adaptive immunity. In tumors immunized with λ3, necrosis was detected along with the appearance of CD3+ and CD8+ lymphocytes specific to the ASPH antigen.

The aggressive nature of HCC makes it a great candidate for gene therapy experiments. Sittiju et al.’s study in 2024 [63] used a hybrid M13 filamentous phage called TPA (containing inverted terminal repeats from the adeno-associated virus) to deliver the TRAIL transgene into Huh-7 and HepG2 HCC cell lines. Although TRAIL is a promising alternative to chemotherapy and radiation therapy, it is cleared from blood circulation quickly; thus, a form of TRAIL treatment that does not require repeated administration would be ideal, such as through a genetic approach. The experiment showed successful delivery of the tmTRAIL gene by binding the RGD4C ligand (displayed on phage pIII) to α_v_β_3_ and α_v_β_5_ integrins on the cell membrane. Transduced cells produced tmTRAIL and showed a significant increase in apoptotic markers with induced cell death. More importantly, high selectivity was shown as the TRAIL transgene was not detected in healthy liver cells.

Chang et al.’s 2016 study [64] developed a delivery system using MS2 virus-like particles (VLPs) crosslinked with the GE11 polypeptide. MS VLPs delivered long non-coding RNA maternally expressed gene 3 (*MEG3*) specifically to epidermal growth factor receptor (EGFR)-positive hepatocellular carcinoma (HCC) cell lines, without activating EGFR downstream signaling pathways. This significantly inhibited tumor cell growth (HepG2, Hep3B, Huh7) and invasion (HepG2) both in vitro and in mice, indicating that MS2 VLP is a promising approach for long non-coding RNA -based cancer therapy.

In 2021, Zhang et al. [65] explored the anti-tumor effects of two microRNAs using MS2 VLPs containing the miR-21 sponge and pre-miR-122 sequences. These VLPs were crosslinked with a peptide targeting HCC cells. The VLPs delivered the miR-21 sponge into cells, and the linked pre-miR-122 was processed into mature miR-122. The miR-21 sponge inhibited the proliferation, migration, and invasion of HCC cells. Simultaneous delivery of the miR-21 sponge and miR-122 further suppressed proliferation (34%), migration (63%), and invasion (65%), while promoting apoptosis in HCC cells. These results indicate that MS2 VLPs-delivered microRNAs are also efficient therapeutic approaches for HCC.

### 5.5. B Cell Lymphoma

B cell lymphoma makes up about 85% of non-Hodgkin lymphomas, which are cancerous lymphocytes. Aggressive lymphomas are treated with chemotherapy, while indolent lymphomas can be treated with radiation in earlier stages [66]. Relapse is a major challenge that scientists are trying to overcome; thus researchers are still experimenting with other treatment methods such as anti-idiotype vaccinations. The goal of anti-idiotype vaccinations is to generate immunogenicity of the tumor-specific antigen idiotype, Id, which is expressed on B cell cancers. The idea of generating anti-idiotype antibodies has been proposed since the late 1980s [67]. The current gold standard of idiotype vaccines is to chemically link idiotype to immunogenic protein keyhole limpet hemocyanin (KLH) [68], but recently there are studies where idiotype is being linked to the M13 phage. In Roehnisch et al.’s study [69], an anti-idiotype vaccination featuring idiotype chemically linked to pVIII on M13 phage was administered to BALB/c mice with the BCL1 lymphoma cell model. Mice vaccinated with the experimental phage survived around 20 days longer than mice vaccinated with wild type phage or buffer. This method produced higher BCL1 specific IgG levels compared to idiotype chemically linked to KLH or idiotype recombinantly expressed on pVIII, demonstrating the benefits of using phage as carriers of specific antigens.

### 5.6. Multiple Myeloma

Multiple myeloma (MM) is characterized by cancerous plasma cells in the bone marrow that can cause anemia, bone damage, and kidney injury. Standard induction therapy is usually a combination of a proteasome inhibitor, immunomodulatory agent, and corticosteroid. Stem cell transplant is always an option if patients are eligible [70]. Scientists are also exploring the anti-idiotype vaccine idea to provide better treatment outcomes for MM.

In Roehnisch et al.’study [68], a vaccine containing engineered M13 phage that was chemically linked to the antigen idiotype was administered to 15 patients with multiple myeloma. Four out of five patients in the 1.25 mg phage dosage treatment group had an increase in anti-Id antibodies and some showed a decrease or stabilization of paraprotein levels. This version of the vaccine, with chemically linked idiotype, proved to be more effective than the control where idiotype was chemically linked to (KLH), as anti-phage antibodies were raised more quickly than anti-KLH antibodies. Since KLH-linked idiotype is considered a standard for idiotype vaccinations, this improved reaction efficiency with phages could suggest a new approach to explore and refine. However, due to the small sample size of this study, more research is needed to confirm treatment efficiency.

### 5.7. Cervical Cancer

Human papillomavirus (HPV) can lead to abnormal growth of cells in the mucosa lining the cervix. If the growth stays long enough, it can eventually lead to cervical cancer [71]. Primary treatments are surgery to remove parts of or the entire uterus, or radiation therapy which is sometimes used in conjunction with chemotherapy [72]. Specific targeting of cancerous cells is an important consideration when treating cervical cancer. 

Current targeted therapies leverage HPV oncoproteins that are expressed in tumor cells as target antigens for vaccines. In one study, HPV E7 was used as a target antigen in a vaccine designed to treat C57BL/6 mice with TC-1 cells. The nanoparticle was a recombinant λ ZAP cytomegalovirus vector with a HPV16 *E7* gene inserted to allow expression of HPV E7 on the surface. Lymphocytes in mice vaccinated with λ ZAP E7 phage had increased specific cytolytic activity with greater effector-to-target ratio compared to wild type phage and negative control (buffer solution), as measured through an LDH release assay. Splenocytes showed the release of interferon-g and granzyme B, demonstrating an immune response. Overall tumor growth was reduced after three injections in one week [73].

### 5.8. Neuroendocrine Pancreatic Tumor

Neuroendocrine tumors (NETs) result from abnormal neuroendocrine cell growth and can arise in the gastrointestinal tract (48%), lung (25%), pancreas (9%), and other organs such as breast, prostate, thymus, and skin [74]. For pancreatic NETs, surgical resection of malignant tissue is currently the only reliable treatment option. However, due to the rarity and slow growth of this disease, most patients present with advanced or metastasized tumors by the time of diagnosis. At late metastatic stages, surgery is no longer reliable, leaving only a few options such as chemotherapy, targeted therapy, and radiation therapy to reduce tumor growth. Somatostatin analogs like octreotide can bind to somatostatin receptors (which are overexpressed in neuroendocrine tissues) to inhibit tumor growth, but drug resistance is still a challenging problem [75].

To capitalize on the known ligand and somatostatin receptor interaction, Smith et al.’s study [76] designed a hybrid particle termed AAVP (combining M13 phage and adeno-associated virus) that displays an octreotide motif to bind to somatostatin receptor type 2 (SSTR2) and deliver a tumor necrosis factor (*TNF*) gene. Mice (Pdc1-Cre) with the gene mutation in multiple endocrine neoplasia syndrome type 1 (*MEN1*) were intravenously administered the octreotide-AAVP-TNF or the control untargeted AAVP-TNF. After four days, TNF expression was detected in the pancreas of mice treated with octreotide-AAVP-TNF and not in other organs or in mice treated with the control, demonstrating targeting specificity and successful transduction. Findings from the 28-day study showed that tumor size and choline levels (markers of malignant tumors) decreased in mice treated with octreotide-AAVP-TNF while both factors increased in mice treated with the control (after single administration). The long-term 2-month study (with weekly administration) observed mice treated with octreotide-AAVP-TNF, and it showed decreased secretion of serum insulin compared to mice treated with the control. This result yields clinical significance as many patients with NETs in the pancreas experience hormonal hypersecretion.

### 5.9. Melanoma

Melanoma is a deadly form of skin cancer with a high risk of metastasis that arises from melanocytes. Surgical resection is the main treatment for melanoma, but if metastasis has occurred, subsequent treatment with chemotherapy, targeted therapies, or immune therapy is also administered. Drug resistance and a need for less toxic drugs drive research in melanoma treatment [77].

A 2024 study from Brišar et al. [78] tested the extent of adaptive immunity activation by a M13 phage-based vaccine expressing tumor peptide MAGE-A1. Two forms of engineered M13 phage were tested, with MAGE-A1 fused to coat protein pVIII or pIII. Mice were vaccinated every two weeks for three rounds with either experimental form while control groups received wild-type M13. Mice with either phage treatment showed greater levels of anti-MAGE-A1 antibodies, with the levels increasing after each administered dose. Through in vitro experiments, it was observed that anti-MAGE-A1 antibodies could bind to melanoma cells from the B16F10 cell line. Furthermore, immune cells of mice vaccinated with either form of the vaccine showed increased cytotoxic activity as B16F10 cells that were incubated with mice’s splenocytes had a lower survival rate than the control.

Shukla et al. in 2018 [79] reported the preclinical study of immunization with T7 phage nanoparticles that display murine melanoma B16F10 neoantigens to elicit plasma antibody and vaccine-draining lymph node B cell responses. They found that a single injection of T7 peptide vaccine was able to stimulate an anti-peptide response, and shorter peptides with 11 amino acids stimulated antibodies to the mutated epitope better than longer peptides with 34 amino acids. A trimer of 11 repeated amino acids provided no advantage over a single monomer as a T7 expressed vaccine.

Hwang and Myung et al.’s study [80] in 2020 tested T7 phage displaying a peptide able to target B16F10 cells. This engineered T7 phage contained a mammalian expression cassette of the cytokine granulocyte macrophage-colony stimulating factor (GM-CSF). GM-CSF was expressed in the transduced cells in vitro and in vivo. Mice treated with the phage intravenously survived for 25 days, whereas only 40% of untreated mice survived. Over the 16 days of treatment, this engineered T7 phage reduced tumor growth by 72% compared to untreated controls. Serum levels of IL-1α, TNF-α, and GM-CSF increased during treatment. By altering the microenvironment and promoting the recruitment of anti-tumor immune cells (macrophages, dendritic cells (DCs), and CD8+ T cells), recombinant T7 phage effectively inhibited tumor growth.

In 2023, Rashidijahanabad et al. [81] reported a novel anticancer vaccine by covalently conjugating sialyl Lewis^a^, an attractive carbohydrate-associated cancer antigen to bacteriophage Qβ. The sera from SLewis^a^-Qβ immunized mice exhibited higher anti-SLewisa antibody titers, which remained detectable at high levels even after 379 days. C57BL/6 mice were immunized with Qβ-sLe^a^ conjugate or control (Qβ + sLe^a^ admixture), and injected with B16 melanoma cells stably expressing fucosyltransferase 3 (B16FUT3, which led to cell surface production of sLe^a^). The mice vaccinated with Qβ-sLe^a^ conjugate had significantly lower numbers of tumor foci in the lungs compared to the group administered with the control.

### 5.10. Chondrosarcoma

Chondrosarcoma is the second most common type of primary bone cancer that presents as cartilaginous neoplasms [82]. It primarily affects the cartilage cells of extremities (45%) followed by those of the axial skeleton (31%) [83]. Chemotherapy and radiation therapy for chondrosarcoma are ineffective due to the slower growth, low division rate, and poor drug penetration; thus, surgery, including surgical excision or amputations, is the main treatment option [84].

Chongchai et al.’s study in 2024 [85] also employed TRAIL-based gene therapy by delivering the sTRAIL transgene (secreted form of TRAIL) to target tumor cells, only the study explored the effect on chondrosarcoma. A phage-derived particle (PDP) was created to serve as the vehicle for gene delivery, consisting of M13 f1 origin of replication and adeno-associated virus (AAV2)-based transgene cassette. To initiate transduction, the RGD4C ligand (attached to pIII) bound to α_v_β_3_ and α_v_β_5_ integrins on the cell membrane. An in vitro experiment showed that PDP successfully delivered the transgene to SW1353 CS cells but not to normal chondrocytes, demonstrating effective selectivity. CS cells with sTRAIl delivery not only had high transcriptional expression of sTRAIL but also showed increased expression of caspases 3 and 8 and decreased expression of genes *XIAP* and *cFLIP* (apoptosis inhibitors), thus confirming that sTRAIL triggers apoptosis. When mice with human CS were administered the RGD4C-expressing PDP, they exhibited decreased tumor size and viability, while control mice that were administered with PDP without the ligand showed increased size and viability.

### 5.11. Glioblastoma

Glioblastoma (GBM), a common malignant form of brain tumors, has very poor prognosis with a median overall survival of 15 months. Current treatment involves surgery followed up with adjuvant radiation therapy or chemotherapy. Temozolomide (TMZ) is a standard chemotherapy administered to GBM patients, although chemoresistance proves to be a significant obstacle to successful TMZ treatment. Phage-based therapy for GBM is promising since phages can cross the blood brain barrier (BBB) and carry low toxicity, which are features that many other nanoparticles lack [86].

To test the idea of phage-based gene delivery, Przystal et al.’s study [87] designed a hybrid particle featuring an adeno-associated virus genome carrying a Grp78 (a chaperone heat shock protein) promoter. The genetic material was encased in M13 phage capsid displaying RGD4C ligand which targets the particle to α_v_β_3_ and α_v_β_5_ integrins. In vitro experiments show that administration of TMZ after particle entry into the cell boosts Grp78 promoter activity through activation of the unfolded protein response stress pathway: this result was leveraged to increase gene expression. In the presence of the prodrug ganciclovir (GCV), administration of RGD4C/AAVP-Grp78-HSVtk followed by TMZ showed greater in vitro tumor cell destruction of glioblastoma cells (from LN229, U87, and SNB19 cell lines) than solely phage therapy or TMZ therapy. The same treatment combination as was used on mice with intracranial tumors from U87 cells and showed a reduction in tumor size and viability.

**Table 2 ijms-25-09938-t002:** Studies of phage-based therapies in different cancers.

	Phage	Peptide Display	Mechanism	Result	Year
Breast cancer (BALB/*c* mice)	T7 phage	Hsp 27	Induction of antibodies	Significantly developed smaller average tumor weight (1.5 ± 0.2 g vs. 2.5 ± 0.5 g) in mice, which were subcutaneously injected with 4T1 breast carcinoma cell.	2008 [49]
Lung carcinoma (mice)	T4 phage	mFlt4	Induction of antibodies	Prolonged survival in mice, which were injected with Lewis lung carcinoma cells	2009 [57]
Lung carcinoma (C57BL/6J mice)	T4 phage	mVEGFR2	Induction of antibodies	Obvious inhibition of tumor growth and significant extension of survival length	2011 [58]
Cancer associated with human HPV type 16 (C57BL/6 mice)	λ- phage	HPV E7	Induction of cytotoxic immune response	Significant inhibition of tumor growth in mice, which were subcutaneously injected TC-1 cells	2011 [73]
Breast cancer (BALB/*c* mice)	T7 phage	H-2k(d)-restricted CTL epitode (derived from rat HER2/neu)	Induction of cytotoxic immune response	Protected mice against HER2 positive tumor challenge in both prophylactic and therapeutic setting	2012 [50]
B-cell lymphoma (BALB/*c* mice)	M13 phage (pVIII)	Chemically fuse single-chain variable fragment-BCL1	Induction of antibodies	Induced B cell lymphoma Idiotype- specific IgG leads to significant survival benefit in the murine B cell lymphoma	2013 [69]
Multiple myeloma (clinical phase I/II trial, 15 patients)	M13 phage	Chemically link patient-specific purified paraprotein	Induction of idiotype-specific antibodies	Decrease or stabilize paraprotein level in patients	2014 [68]
Hepatocellular Carcinoma (BALB/*c* mice)	MS2 phage	GE11 peptide	Introducing IncRNA MEG3 transgene into EGFR positive HCC cell lines	Significant inhibition of tumor growth	2016 [64]
Pancreatic neuroendocrime tumor (*Men1* tumor suppressor gene KO mice)	Hybrid AAV/phage	Octreotide peptide (binding to SSTR2 on cancer cell)	Introducing TNF transgene directly into cancer cell	Induced apoptosis of cancer cell resulting in lowered tumor metabolism, reduced tumor size (73 ± 21%) and improved mice survival	2016 [76]
Hepatocellular carcinoma (mice)	λ-phage (gp D)	Human Aspartate β-hydroxylase (ASPH) derived proteins	Induction of cytotoxic immune response	Significantly delayed HCC growth and progression in mice, which had subcutaneous implantation of a syngeneic BNL HCC cell lines	2017 [62]
Breast cancer (BALB/*c* mice)	λF7 phage (gpD)	AE37 (Ii-Key/Her-2/neu 776–790)	Induction of cytotoxic immune response	Promising prophylactic and therapeutic effects against HER2 overexpressing cancer in BALB/*c* mice	2018 [46]
Melanoma (mice)	T7 phage	11-AA and 34-AA peptides	Induction of antibodies	11-AA peptides showed better antibody stimulation than longer peptides (34-AA peptides)	2018 [79]
Lung carcinoma (BALB/*c* or C57BL/6 mice)	T4 phage	VEGFR2	VEGFR2 delivery (binding to VEGF), altering tumor microenvironment	Suppress tumor growth and decrease microvascular density in murine models of Lewis lung carcinoma	2019 [59]
Glioblastoma (immunodeficient nude mice)	Hybrid AAV/phage	RGD4C ligand (binding to α_v_β_3_ and α_v_β_5_ on cancer cell)	Introducing *Grp78*-*HSVtk* transgene directly into cancer cell to activate ganciclovir (pro-drug) by phosphorylation	RGD4C/AAVP-*Grp78*-*HSVtk* plus ganciclovir inhibits tumor growth, and efficacy is boosted by Temozolomide	2019 [87]
Lung Cancer (A549 human lung adenocarcinoma cells)	Hybrid AAV2/phage	RGD4C ligand (binding to α_v_β_3_ and α_v_β_5_ on cancer cell)	Tumor suppressor *p53* gene replacement	Bacteriophage vector is efficiently and selectively delivered CRISPR-Cas9 transgene to A549 human lung adenocarcinoma cells	2020 [56]
Melanoma (BALB/*c* mice)	T7 phage	pep42	Introducing expression cassette of GM-CSF transgene into cancer cells	Inhibited tumor growth (B16F10 melanoma cells) by 72% compared to untreated control	2020 [80]
Colorectal cancer (BALB/*c* mice)	M13 phage (pVIII)	GM-CSF (a potent activator for STAT5 signaling in murine macrophage)	Antitumor cytokine therapy, altering tumor microenvironment	Significantly reduced the tumor size (>50%) in mice, which had subcutaneously injected murine CRC cancer cell line CT26	2021 [38]
Hepatocellular Carcinoma (HCC cell line)	MS2 phage	GE-11 peptide	Introducing microRNA-21-sponge and pre-microRNA-122 transgenes into HCC cells	Decreased proliferation, migration and invasion of HCC cells	2021 [65]
Breast cancer (△16HER2 transgenic mice)	M13 phage (pIII)	△16HER2	Induction of anti-HER2 antibody	Significantly delayed tumor onset and reduced tumor growth rate in △16HER2 transgenic mice	2022 [48]
Melanoma (mice)	Qβ phage	Sialyl Lewis antigen	Induction of antibodies	Significantly reduced tumor development in mice metastatic cancer model	2023 [81]
Colorectal cancer (colon cancer cell line)	M13 phage (pIII) conjugated with photosensitizer Rose Bengal	Disulfide-constrained peptide nonamer (CPIEDRPMC)	Photodynamic anticancer effect	Colon cancer cell line (HT29) showed prominent decrease in cell viability	2024 [52]
Hepatocellular carinoma (human HCC cell lines)	Hybrid AAV/phage	RGD4C ligand (binding to α_v_β_3_ and α_v_β_5_ on cancer cell)	Introducing TRAIL gene directly into cancer cell	Induced apoptosis in human HCC cell lines (Huh-7 and HepG2)	2024 [63]
Malignant melanoma (female C57BL/6NCrl mice)	M13 phage (pIII or pVIII)	MAGE-A1 peptide	Induction of antibodies (anti-MAGE-A1 antibody) and cytotoxic immune response	Anti-MAGE-A1 antibody exhibited a binding capability to B16F10 tumor cells in vitro. Splenocytes demonstrated enhanced CTL cytotoxicity against B16F10 cells	2024 [78]
Chondrosarcoma (BALB/*c nu*/*nu* mice)	Hybrid AAV2/phage	RGD4C ligand (binding to α_v_β_3_ and α_v_β_5_ on cancer cell)	Introducing human sTRAIL transgene directly into cancer cell	Decreased in tumor size mediated by tumor cell apoptosis (measured by tumor luminescence values as fold change) in BALB/*c nu*/*nu* mice implanted with human chondrosarcoma cells (SW1353)	2024 [85]

## 6. Discussion

The studies featured in this review were selected to represent the most recent advancements in phage-based therapies for various cancer types. The development of phage-based therapies for oncology is at varying stages. Most phage-based therapies are still being tested on murine models. Although therapies for very few cancer types (such as myeloma) have advanced into human clinical trials, they still lack groundbreaking success. Studies on some cancers, particularly lymphoma, cervical cancer, and myeloma have not been continued since more than a decade ago. On the contrary, cancers such as colorectal cancer, HCC, melanoma, and chondrosarcoma all had studies with experimental phage-based therapy published very recently. Cancers at the forefront of phage-based therapy research are those with more serious drug resistance, where patients have extremely limited treatment options, or that are more well-researched, allowing easier experimental design. It is encouraging to see that multiple mechanisms are being tested on the same cancer type, such as with HCC.

The popularity of M13 phage use for experimental design among other phage types can be attributed to the awarding of the 2018 Nobel Prize in Chemistry for the innovation of the phage display technique. As M13 phage is the most frequently used phage for constructing phage display libraries [88], it is no surprise that many phage-based anti-cancer studies also used M13.

While the growing interest in phage-based cancer therapies engenders hope, there are limitations that must be overcome for successful therapy. First are challenges that phage vectors face during circulation. The small nature of phages can trigger processing by the reticuloendothelial system (RES), which consists of phagocytes that phagocytose foreign particles [89]. Once the phages are cleared by the RES, they are destroyed by macrophages in the spleen or liver, decreasing their half-life. However, there are certain phage variants that can be used, termed long-circulating phages, which can circumvent the RES system and remain in blood circulation for longer (because of a mutation in the capsid) [7,17]. There may be impaired diffusion of vectors, especially into cores of solid tumors, due to high interstitial pressure or the unconventional features of angiogenic vessels; however, vascular-modulating drugs can be administered before phage treatment to overcome these issues [90]. Therapies against multidrug resistant tumors must also bypass the efflux systems that pump anticancer drugs from their cytosol to prevent treatment failure: further research should be conducted to determine vectors that can bypass this system [91]. To reduce challenges during the engineering process of phage constructs, antigen size and phage genome length should be kept reasonable to ensure viable phage particles. Lastly, mutations in receptors could render phage therapy ineffective by disrupting antibody-receptor interaction and losing selectivity. For instance, HER-2 mutations in breast cancer have led to resistance towards drugs such as trastuzumab due to alterations in HER-2 structure [92]. However, it is possible to engineer phages with antigens of known variants to overcome this challenge.

While common phage-based treatment mechanisms include gene therapy and immunotherapy, phage therapy could also be applied to targeting tumor-associated macrophages (TAMs). TAMs have gained scientific attention in recent years for being a promising target for cancer treatment. TAMs can either be M1 macrophages, which are involved in anti-tumor activity including inflammatory responses and antibody-dependent cellular cytotoxicity, or M2 macrophages, which promote tumor progression while inhibiting immune responses [93]. Through molecular interactions, M2 TAMs are also involved in the crucial steps of tumor metastasis including primary site invasion, intravasation, survival of circulating tumor cells, extravasation, and growth in secondary metastatic sites. The cell can change between M1 and M2 phenotypes depending on changes in the tumor microenvironment or other biological cues. Multiple mechanisms to target M2 TAMs can include but are not limited to their elimination (through host defense system or gene suicide), impairment of movement, or prevention of M1 to M2 polarization [94]. TAM research in general is still in its initial stages and additional research is needed to fully understand TAM functions. Although there have been several targeted therapies that have worked on animal models by blocking pathways and intracellular interactions, human patients often respond poorly [95], mostly due to lack of M2-specificity leading to toxicity for healthy cells [96]. With their selectivity and manipulability, phages could serve as efficient vectors for TAM-based treatment. In fact, Cieslewicz et al.’s study reported the discovery of a peptide M2pep that can preferentially target M2 macrophages with low impact on healthy leukocytes in murine models [97]. Although various nanoparticles are currently being developed to increase targeting specificity [95], phages should be taken into greater consideration for their easy and low cost of production.

It is important to understand that while the development of phage-based therapies for certain cancers is gaining momentum, different national government regulations create substantial limitations towards their use and standardization in the real-world context. Several countries are at varying stages of integrating phage therapy into their medical treatments offered. Except for Russia and Georgia, where phage cocktails such as “Pyophage” and “Intestiphage” to treat bacterial infection are available for consumer purchase without prescription [98], most people access phage therapy through compassionate use, which is when seriously ill patients can be treated with an investigational product if there are no clinical trials or safer options available [99]. Countries such as the UK, France, Belgium, Australia, India, China, and the US allow the use of compassionate treatment for severe resistant infectious disease. Another administration of phage-based treatment is through magistral preparation, a personalized approach where the final product is compounded in a pharmacy based on a prescription: this can be seen in Poland, Georgia, and Belgium [100,101]. Although there is no FDA approved phage therapy in the US thus far, the US is making significant strides in phage research, with the most phage-related investigator-sponsored trials (IST) compared to other countries (many of these trials have advanced to phase 3 status) [101]. While countries like India are without clear regulations, most countries are beginning to acknowledge the potential of phage therapy and are pushing for official legislation or research. It is evident that the world is starting to embrace the prospects of phage therapy, although its application to cancer treatment has still not been authorized in any country.

To date, gene and vaccine therapy have gained great interest in scientific research for cancer therapy which could be well explored using different vectors. Compared with other vectors, phages stand out for their high specificity, reduced toxicity, and low cost of production. Ongoing in-depth research on phage-based therapy is no doubt worthwhile and marks an exciting turn in the biomedical sciences as we discover novel applications of phage to treat cancer and other health conditions.

## Figures and Tables

**Figure 1 ijms-25-09938-f001:**
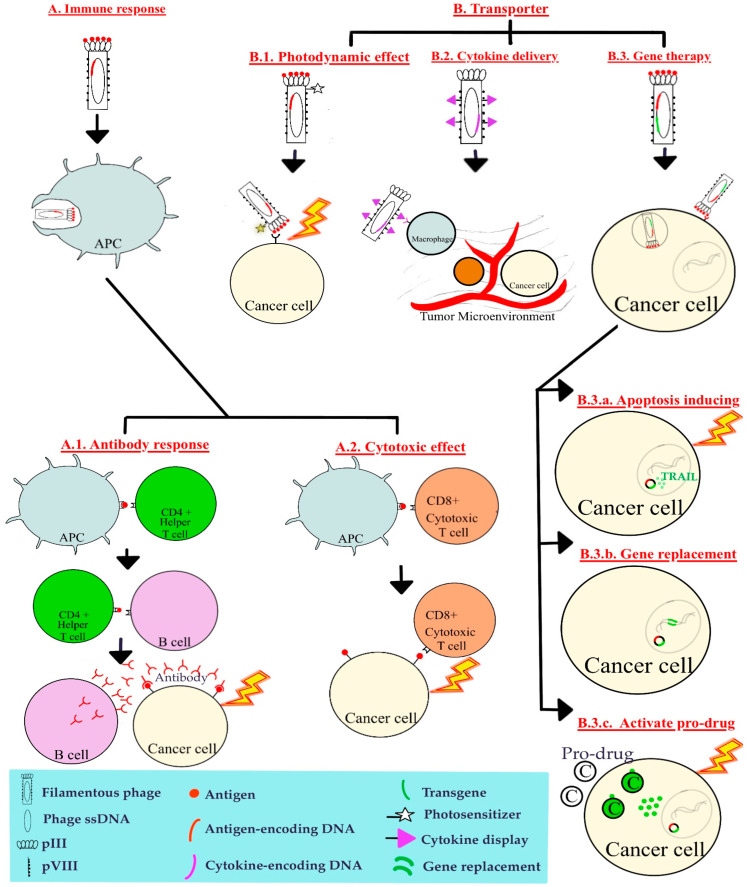
Mechanisms of phage-based cancer therapies (using filamentous phage as an example). (**A**) Cancer cell antigens can be displayed on phage surface using peptide display technique. These antigens can trigger an immune response, encourage production of anti-cancer antibodies, or activate cytotoxic T cells against cancer cells. (**B**) Phages can be used as transporters to deliver photosensitizers, cytokines, or transgenes to cancer cells. APC, antigen presenting cell; TRAIL, tumor necrosis factor-related apoptosis-inducing ligand; pIII, M13 pIII coat protein; pVIII, M13 pVIII coat protein.

**Table 1 ijms-25-09938-t001:** Classification and characterization of bacteriophages used in cancer therapy.

Phage Type	Class	Family	Genus	Genome	Size (kb)	Proteins	Morphology	Life Cycle (in Bacteria)
M13	*Faserviricetes*	*Inoviridae*	*Inovirus*	ssDNA	6.4	11	Filamentous	Lysogenic
λ	*Caudoviricetes*	*Siphoviridae*	*Lambdavirus*	dsDNA	48.5	73	Icosahedral head with tail	Lytic/ lysogenic
T4	*Caudoviricetes*	*Straboviridae*	*Tequatrovirus*	dsDNA	168	289	Icosahedral head with tail	Lytic
T7	*Caudoviricetes*	*Autographiviridae*	*Teseptimavirus*	dsDNA	40	55	Icosahedral head with tail	Lytic
MS2	*Leviviricetes*	*Fiersviridae*	*Emesvirus*	ssRNA	3.6	4	Icosahedral	Lytic
Qβ	*Leviviricetes*	*Fiersviridae*	*Qubevirus*	ssRNA	4.2	4	Icosahedral	Lytic

## Data Availability

The original contributions presented in the study are included in the article, further inquiries can be directed to the corresponding authors.

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
