# Peer review of "Recent Advances and Mechanisms of Phage-Based Therapies in Cancer Treatment"

_ijms, 2024, doi:10.3390/ijms25189938_

Round 1

Reviewer 1 Report

Comments and Suggestions for Authors

The authors summarized the current methods of using phages for cancer treatment. It covers a wide range of cancer types but a quite limited number of phages. This manuscript does not provide much novelty or advantage over other similar reviews, but it still stands scientifically without significant flaws. I think it is a publishable work if these issues could be addressed:

It reads strange when it is stated that most phages are 40-200 nm, but filamentous phages are ‘only’ a few micrometers (the second paragraph of page 2).

I believe Figure 1, the only figure in this article, should be improved with better labeling and annotations. What is the rectangle attached to each cell? If it represents a phage particle, it should be of icosahedron head like the authors stated in the text. What are the dots on the rectangles? I guess they are antigens on the phage surface, but why do some of them have red dots attached while others don’t? What is the difference? What is the circle in the rectangle? I guess it is nucleic acid, but why does it have different colors in different cases? I am only taking the ‘rectangle complex’ as an example, but the fact is the sole element that is properly labeled in this figure is the cell type, so please put more efforts to make an understandable figure.

Table 1 looks informative, but it can be noticed that this table, and of course the entire manuscript, are mainly focused on M13. Can other phages be used for cancer treatment? What about classical T4 and T7? To my limited knowledge, T4 and T7 can also be engineered for gene therapy. It would be really nice if the authors could include more types of phages in this work since it is entitled ‘phage-based therapies’ not ‘M13-based therapies’.

Author Response

Reviewer 1:

The authors summarized the current methods of using phages for cancer treatment. It covers a wide range of cancer types but a quite limited number of phages. This manuscript does not provide much novelty or advantage over other similar reviews, but it still stands scientifically without significant flaws. I think it is a publishable work if these issues could be addressed:

Q1: It reads strange when it is stated that most phages are 40-200 nm, but filamentous phages are ‘only’ a few micrometers (the second paragraph of page 2).

--Ans: We have deleted ‘only’ from the sentence [7].

Q2: I believe Figure 1, the only figure in this article, should be improved with better labeling and annotations. What is the rectangle attached to each cell? If it represents a phage particle, it should be of icosahedron head like the authors stated in the text. What are the dots on the rectangles? I guess they are antigens on the phage surface, but why do some of them have red dots attached while others don’t? What is the difference? What is the circle in the rectangle? I guess it is nucleic acid, but why does it have different colors in different cases? I am only taking the ‘rectangle complex’ as an example, but the fact is the sole element that is properly labeled in this figure is the cell type, so please put more efforts to make an understandable figure.

--Ans: We have added labels (component of phage, prodrug, TRAIL, antibody) and their description in Figure 1: “Figure 1. Mechanisms of phage-based cancer therapies (using filamentous phage as an example). (A) Cancer cell antigens can be displayed on phage surface using peptide display technique. These antigens can trigger an immune response, encourage production of anti-cancer antibodies, or activate cytotoxic T cells against cancer cells. (B) Phages can be used as transporters to deliver photosensitizers, cytokines or transgenes to cancer cells. APC, antigen presenting cell; TRAIL, tumor necrosis factor-related apoptosis-inducing ligand.

Q3: Table 1 looks informative, but it can be noticed that this table, and of course the entire manuscript, are mainly focused on M13. Can other phages be used for cancer treatment? What about classical T4 and T7? To my limited knowledge, T4 and T7 can also be engineered for gene therapy. It would be really nice if the authors could include more types of phages in this work since it is entitled ‘phage-based therapies’ not ‘M13-based therapies’.

--Ans: We have added more types of phages studies below, and updated Table 2.

5.1 Bresast cancer:

In 2008, Shadidi et al. showed that oral immunization with T7 phage displaying the tumor antigen Hsp27 could lead to the induction of effective immune responses in 4T1 murine breast cancer cells which overexpress Hsp27. Mice were subcutaneously injected with 4T1 cells on day 30 after immunization with the T7 phage expressing the Hsp27. Mice orally immunized with T7 phage expressing the Hsp27 developed tumors that had a significantly smaller average weight versus the wild-type phage (1.5±0.2g versus 2.5±0.5g; p<0.05). Additionally, mice immunized with the wild-type phage had significantly higher number of lung metastases than Hsp27 phage treatment (80±15 vs 40±10, p< 0.05). These data provided the first evidence that a recombinant T7 phage stably expressing Hsp27 can inhibit outgrowth of 4T1 breast cancer cells [49].

In Pouyanfard et al.’s 2012 study, multivalent T7 bacteriophage nanoparticles were developed that displayed an immunodominant H-2kd-restricted CTL epitope (9-mer  peptide p66 TYVPANASL) derived from the rat HER2/neu. The authors tested the therapeutic potential of these chimeric T7 nanoparticles in 6- to 8-week-old female BALB/c mice. Only T7 phage nanoparticles carrying a single copy of p66 peptide were successfully cross-presented and elicited a specific T cell response against the displayed CTL epitope. Immunization with these nanoparticles induced CTL-mediated lysis of P815 target cells in vitro and increased IL-4 cytokine secretion. T7-p66-vaccinated mice efficiently rejected HER2-expressing TUBO Cells. Moreover, anti-tumor effects and dramatic tumor regression were significant in some mice vaccinated with p66+FA, whereas all control mice developed large tumors by day eighty [50].

5.3 Lung adenocarcinoma

In Ren et al.’s study, a T4 phage nanoparticle expressing mouse Flt4 (mFlt4), which promotes tumor metastasis by stimulating solid tumor lymphangiogenesis, was constructed to evaluate the phage's antitumor activity. The effect of T4-mFlt4 vaccine was explored in mice injected with cells of Lewis lung carcinoma (LLC), a VEGFR-positive tumor. Mice with LLC-derived tumors showed extended survival when treated with the T4-mFlt4 vaccine compared to the control group. The treated group had a survival duration of 64 days, compared to 29 and 33 days in the control groups. By day 62, the survival rate in the T4-mFlt4 group was 25%, while the control phage group had a survival rate of 12.5% at 29 days. The vaccine did not demonstrate an ability to inhibit tumor growth, but it suppressed tumor metastasis (0.45 g±0.12 g versus 0.84 g±0.29 g in the   control, P=0.002) in mice. Histochemical examinations showed that the mean lymphatic microvessel counts were reduced in tumors after T4-mFlt4 treatment, but the average vascular microvessel counts in lungs were not significantly different between the groups [57].

Ren et al.’s 2011 study demonstrated the protective immunity of T4-mVEGFR2   vaccine against Lewis lung carcinoma (LLC) in mice. In vitro studies showed that immunoglobulin induced by T4-mVEGFR2 inhibited VEGF-mediated endothelial cell proliferation and tube formation. The antitumor activity was further confirmed through adoptive transfer of the purified immunoglobulin. The antitumor activity and production of autoantibodies against mVEGFR2 were neutralized when CD4+ T lymphocytes were depleted [58].

Zuo et al.’s 2019 study evaluated the anti-angiogenic effects of recombinant T4 phages expressing the extracellular domain of VEGFR2 (T4-VEGFR2). The T4-VEGFR2 phages specifically bound to VEGF and inhibited VEGF-induced phosphorylation of VEGFR2 and downstream kinases, such as extracellular signal-regulated kinase (ERK) and p38 mitogen-activated protein kinase (MAPK). In vitro experiments demonstrated that T4-VEGFR2 phages could inhibit VEGF-induced endothelial cell proliferation and migration. The administration of T4-VEGFR2 phages suppressed tumor growth, reduced microvascular density, and prolonged survival in murine models of LLC and colon    carcinoma (CT26 cell line) [59].

5.4 Hepatocellular carcinoma

Chang et al.’s 2016 study developed a delivery system using MS2 virus-like particles (VLPs) crosslinked with the GE11 polypeptide. MS VLPs delivered long non-coding RNA maternally expressed gene 3 (MEG3) specifically to epidermal growth factor receptor (EGFR)-positive hepatocellular carcinoma (HCC) cell lines, without activating EGFR downstream signaling pathways. It significantly inhibited tumor cell growth (HepG2, Hep3B, Huh7) and invasion (HepG2) both in vitro and in mice, indicating that MS2 VLP is a promising approach for long non-coding RNA -based cancer therapy [64].

In 2021, Zhang et al. explored the anti-tumor effects of two microRNAs using MS2 VLPs containing the miR-21 sponge and pre-miR-122 sequences. These VLPs were crosslinked with a peptide targeting HCC cells. The VLPs delivered miR-21 sponge into cells, and the linked pre-miR-122 was processed into mature miR-122. The miR-21 sponge inhibited the proliferation, migration, and invasion of HCC cells. Simultaneous delivery of miR-21 sponge and miR-122 further suppressed proliferation (34%), migration (63%), and invasion (65%), while promoting apoptosis in HCC cells. These results indicate that MS2 VLPs-delivered microRNAs are also efficient therapeutic approaches for HCC [65].

5.9 Melanoma

Shukla et al. in 2018 reported the preclinical study of immunization with T7 phage nanoparticles that display murine melanoma B16F10 neoantigens to elicit plasma antibody and vaccine-draining lymph node B cell responses. They found that a single   injection of T7 peptide vaccine was able to stimulate an anti-peptide response, and shorter peptides with 11 amino acids stimulated antibodies to the mutated epitope better than longer peptides with 34 amino acids. A trimer of 11 repeated amino acids provided no advantage over single monomer as a T7 expressed vaccine [79].

Hwang and Myung et al.’s study in 2020 tested T7 phage displaying a peptide able to target B16F10 cells. This engineered T7 phage contained a mammalian expression cassette of the cytokine granulocyte macrophage-colony stimulating factor (GM-CSF). GM-CSF was expressed in the transduced cells in vitro and in vivo. Mice treated with the phage intravenously survived for 25 days, whereas only 40% of untreated mice survived. Over the 16 days of treatment, this engineered T7 phage reduced tumor growth by 72% compared to untreated controls. Serum levels of IL-1α, TNF-α, and GM-CSF increased during treatment. By altering the microenvironment and promoting the recruitment of anti-tumor immune cells (macrophages, dendritic cells (DCs), and CD8+ T cells), recombinant T7 phage effectively inhibited tumor growth [80].

In 2023, Rashidijahanabad et al. reported a novel anticancer vaccine by covalently conjugating sialyl Lewisa, an attractive carbohydrate-associated cancer antigen to bacteriophage Qβ. The sera from SLewisa-Qβ immunized mice exhibited higher anti-SLewisa antibody titers, which remained detectable at high levels even after 379 days. C57BL/6 mice were immunized with Qβ-sLea conjugate or control (Qβ+sLea admixture), and injected with B16 melanoma cells stably expressing fucosyltransferase 3 (B16FUT3, which led to cell surface production of sLea). The mice vaccinated with Qβ-sLea conjugate had significantly lower numbers of tumor foci in lungs compared to the group         administered with the control [81].

Table 2.

phage

Peptide display

Mechanism

Result

Year

Breast cancer

(BALB/c mice)

T7 phage

Hsp 27

Induction of antibodies

Significantly developed smaller average tumor weight (1.5± 0.2g vs 2.5± 0.5g) in mice, which were

subcutaneously injected with 4T1 breast carcinoma cell.

2008 [49]

Lung carcinoma

(mice)

T4 phage

mFlt4

Induction of antibodies

Prolonged survival in mice, which were injected with Lewis lung

carcinoma cells

2009 [57]

Lung carcinoma

(C57BL/6J mice)

T4 phage

mVEGFR2

Induction of antibodies

Obvious inhibition of tumor growth and significant extension of survival length

2011 [58]

Breast cancer

(BALB/c mice)

T7 phage

H-2k(d)-restricted CTL epitode

(derived from rat HER2/neu)

Induction of cytotoxic

immune response

Protected mice against HER2

positive tumor challenge in both prophylactic and therapeutic

setting

2012 [50]

Hepatocellular

Carcinoma

(BALB/c mice)

MS2 phage

GE11 peptide

Introducing IncRNA MEG3  transgene into EGFR positive HCC cell lines

Significant inhibition of

tumor growth

2016 [64]

Lung carcinoma

(BALB/c or C57BL/6 mice)

T4 phage

VEGFR2

VEGFR2 delivery (binding to VEGF), altering tumor microenvironment

Suppress tumor growth and decrease microvascular density in murine models of Lewis lung carcinoma

2018 [59]

Melanoma

(mice)

T7 phage

11-AA and 34-AA peptides

Induction of antibodies

11-AA peptides showed better  antibody stimulation than longer peptides (34-AA peptides)

2018 [79]

Melanoma

(BALB/c mice)

T7 phage

pep42

Introducing expression cassette of GM-CSF transgene into cancer cells

Inhibited tumor growth (B16F10 melanoma cells) by 72% compared to untreated control

2020 [80]

Hepatocellular

Carcinoma

(HCC cell line)

MS2 phage

GE-11 peptide

Introducing microRNA-21-sponge and pre-microRNA-122 transgenes into HCC cells

Decreased proliferation, migration and invasion of HCC cells

2021 [65]

Melanoma

(mice)

Qβ phage

Sialyl Lewis antigen

Induction of antibodies

Significantly reduced tumor development in mice metastatic cancer model

2023 [81]

Reviewer 2 Report

Comments and Suggestions for Authors

The manuscript titled "Current Developments and Mechanisms of Phage-Based Therapies of Different Cancers" offers a thorough exploration of the emerging role of bacteriophages in cancer treatment. The review effectively summarizes recent advancements and highlights the potential of phage-based therapies to address key challenges in oncology, such as drug resistance and the specificity of treatment. While the manuscript is comprehensive and well-researched, there are several areas where further clarification and refinement could enhance the overall quality and readability of the review. Below, I provide detailed comments and suggestions for improvement.

 Specific Comments:

Title:

The title is clear and informative, capturing the essence of the paper. However, it might benefit from a slight rephrasing to make it more precise and impactful. Consider something like: "Recent Advances and Mechanisms of Phage-Based Therapies in Cancer Treatment."

Abstract:

1. The abstract effectively summarizes the main points of the review, highlighting the potential of phage-based therapies in cancer treatment. However, it could be more specific about the types of cancers studied and the key findings from recent research. Including a brief mention of the types of engineered phages or specific mechanisms discussed in the review would also strengthen the abstract.

2. The sentence "This option comes suitably at a time when challenges in cancer research include damage of healthy cells, targeting inefficiency, barrier obstruction, and drug resistance" is a bit long and could be clearer if broken down. For example: "This approach is particularly timely, as current challenges in cancer research include the damage to healthy cells, inefficiency in targeting, obstruction by biological barriers, and drug resistance."

3. The abstract mentions that some cancers have not been revisited in a decade, but it would be helpful to specify which types of cancers this refers to, or at least give an example. This would add more context to the review’s scope and relevance.

1. Introduction

 1. The introduction effectively sets the stage by highlighting the ongoing challenges in cancer treatment and the need for innovative therapies. The statistic that "1 in 5 people develop cancer in their lifetime" is compelling and underscores the urgency of the issue. However, consider specifying whether this statistic refers to a global estimate or is specific to a particular region or population.

2. The definition of cancer and its complexities are well articulated. To enhance clarity, consider rephrasing the sentence: "The sheer complexity of cancer, specifically the differences in molecular and genetic behavior across subtypes, renders it difficult to treat and cure completely." It could be more concise, such as: "The complexity of cancer, particularly the molecular and genetic variability across its subtypes, makes it challenging to treat and ultimately cure."

2. What are bacteriophages?

1. The historical background provided is informative and relevant, giving the reader insight into the discovery and early research on phages. However, the transition from the historical context to the modern understanding of phages could be smoother. Consider connecting these ideas by stating how early discoveries laid the foundation for current applications.

2. The section does a good job of describing the physical characteristics and structure of phages. The mention of their size and structural components is important, but it might be helpful to briefly explain why these features make phages suitable for therapeutic use.

3. The description of the lytic and lysogenic cycles is well done, providing clear distinctions between virulent and temperate phages. To tie this back to the theme of cancer therapy, consider mentioning how these cycles can be leveraged in therapeutic applications.

3. Phage-base therapy

1. The advantages of phage-based therapies over conventional cancer treatments are well articulated, particularly the specificity, reduced toxicity, and versatility of phages. It might be helpful to briefly compare these advantages with the limitations of current treatments to strengthen the argument.

2. The potential applications of phage display in cancer therapy are well described, particularly its role in targeted delivery and immune response activation. To reinforce the significance of this technology, consider adding a sentence on how phage display could overcome current challenges in cancer treatment.

4. Main mechanisms of phage-based therapies

1. Consider briefly mentioning the potential advantages of using phages in this context, such as the ability to present a wide array of tumor-specific antigens or the potential for personalized cancer vaccines.

2. The section on gene therapy is detailed and provides a good overview of different therapeutic strategies, including apoptosis induction, gene replacement, and prodrug activation. However, the text could be made more engaging by highlighting the potential impact of these strategies on patient outcomes, such as increased survival rates or reduced side effects. Consider discussing some of the challenges or limitations of these approaches, such as potential off-target effects or the need for precise delivery mechanisms, and how phages might help overcome them.

6. Discussion

1. After discussing each challenge, it could be useful to briefly mention current strategies or research directions aimed at overcoming these issues.

2. To reinforce the potential of phage-based therapies in targeting TAMs, you could compare the specificity and manipulability of phages with other TAM-targeting strategies, emphasizing the unique advantages phages might offer.

Comments on the Quality of English Language

Minor editing of English language is required.

Author Response

Reviewer 2:

The manuscript titled "Current Developments and Mechanisms of Phage-Based Therapies of Different Cancers" offers a thorough exploration of the emerging role of bacteriophages in cancer treatment. The review effectively summarizes recent advancements and highlights the potential of phage-based therapies to address key challenges in oncology, such as drug resistance and the specificity of treatment. While the manuscript is comprehensive and well-researched, there are several areas where further clarification and refinement could enhance the overall quality and readability of the review. Below, I provide detailed comments and suggestions for improvement.

 Specific Comments:

Title:

Q1. The title is clear and informative, capturing the essence of the paper. However, it might benefit from a slight rephrasing to make it more precise and impactful. Consider something like: "Recent Advances and Mechanisms of Phage-Based Therapies in Cancer Treatment."
-- Ans: We have changed the title to "Recent advances and mechanisms of phage-based therapies in cancer treatment."

Abstract:

Q1. The abstract effectively summarizes the main points of the review, highlighting the potential of phage-based therapies in cancer treatment. However, it could be more specific about the types of cancers studied and the key findings from recent research. Including a brief mention of the types of engineered phages or specific mechanisms discussed in the review would also strengthen the abstract.

--Ans: We mentioned the types of phages and were clearer on the specific mechanisms discussed (“Some cancers are being kept in the forefront of phage research, such as colorectal cancer and HCC, while others like lymphoma, cervical cancer, and myeloma have not been retouched in a decade. Common mechanisms are immunogenic antigen display on phage coats and the use of phage as transporters to carry drugs, genes, and other molecules. To date, popular phage treatments being tested are gene therapy and phage-based vaccines using M13 and λ phage, with some vaccines having advanced to human clinical trials.”)

Q2. The sentence "This option comes suitably at a time when challenges in cancer research include damage of healthy cells, targeting inefficiency, barrier obstruction, and drug resistance" is a bit long and could be clearer if broken down. For example: "This approach is particularly timely, as current challenges in cancer research include the damage to healthy cells, inefficiency in targeting, obstruction by biological barriers, and drug resistance."

--Ans: We have changed the sentence to the suggested revision.  (“This approach is particularly timely, as current challenges in cancer research include the damage to healthy cells, inefficiency in targeting, obstruction by biological barriers, and drug resistance.")

Q3. The abstract mentions that some cancers have not been revisited in a decade, but it would be helpful to specify which types of cancers this refers to, or at least give an example. This would add more context to the review’s scope and relevance.

--Ans: We have mentioned the specific cancers that have not been revisited in a decade (“Some cancers are being kept in the forefront of phage research, such as colorectal cancer and HCC, while others like lymphoma, cervical cancer, and myeloma have not been retouched in a decade.”)

  1. Introduction

 Q1. The introduction effectively sets the stage by highlighting the ongoing challenges in cancer treatment and the need for innovative therapies. The statistic that "1 in 5 people develop cancer in their lifetime" is compelling and underscores the urgency of the issue. However, consider specifying whether this statistic refers to a global estimate or is specific to a particular region or population.

--Ans: Since this was a global statistic, we restated the sentence to “1 in 5 people in the world develop cancer in their lifetime.

Q2. The definition of cancer and its complexities are well articulated. To enhance clarity, consider rephrasing the sentence: "The sheer complexity of cancer, specifically the differences in molecular and genetic behavior across subtypes, renders it difficult to treat and cure completely." It could be more concise, such as: "The complexity of cancer, particularly the molecular and genetic variability across its subtypes, makes it challenging to treat and ultimately cure."

 --Ans: We have revised the sentence to the suggested revision. (“The complexity of cancer, particularly the molecular and genetic variability across its subtypes, makes it challenging to treat and ultimately cure.”)

  1. What are bacteriophages?

Q1. The historical background provided is informative and relevant, giving the reader insight into the discovery and early research on phages. However, the transition from the historical context to the modern understanding of phages could be smoother. Consider connecting these ideas by stating how early discoveries laid the foundation for current applications.

--Ans: To make the transition from the historical context to our modern understanding, the following sentence has been added: “After years of research, the fundamental properties and mechanics of phages are now better understood. These powerful nanoparticles range in size from 40 to 200nm.”

Q2. The section does a good job of describing the physical characteristics and structure of phages. The mention of their size and structural components is important, but it might be helpful to briefly explain why these features make phages suitable for therapeutic use.

--Ans: To describe why their small size and sound structural shape is important, the sentence after mention of their size has been added: “Since structure is not compromised by their small size, phages are suitable for application in nanomedicine as different structural components can manipulated through bioengineering.”

Q3. The description of the lytic and lysogenic cycles is well done, providing clear distinctions between virulent and temperate phages. To tie this back to the theme of cancer therapy, consider mentioning how these cycles can be leveraged in therapeutic applications.

 --Ans: Since phages are unable to replicate in mammalian cells, the lytic or lysogenic properties of phages for use in cancer treatment are not as strongly considered. 

  1. Phage-base therapy

Q1. The advantages of phage-based therapies over conventional cancer treatments are well articulated, particularly the specificity, reduced toxicity, and versatility of phages. It might be helpful to briefly compare these advantages with the limitations of current treatments to strengthen the argument.

--Ans: To strengthen the argument, we first state the advantages then explain how they overcome limitations of current treatments: “The main advantage is the high specificity of phages for their target host [20].  Current chemotherapy has lower specificity as it destroys all rapidly proliferating cells, regardless of whether they are tumorigenic or healthy [21-22]. Enhanced selectivity can reduce the side effects that are commonly associated with chemotherapeutic agents (such as nausea, vomiting, hair loss, decreased appetite, and bone marrow suppression) [21] by attacking only those cells that display a certain marker and leaving healthy cells alone. Alternatively, a more holistic approach of “training” the immune system to recognize  specific markers on cancerous cells could serve as natural long-term protection against possible metastasis or recurrence. Another advantage is reduced toxicity: phages are mostly made of nucleic acids and proteins, making them inherently non-toxic. This is a major advantage over chemotherapy and radiation therapy, which expose the body to toxic chemicals and rays. Phages are also highly versatile and can carry gene-editing tools alongside the antigen to treat cells at the genetic level, a feature not found in conventional therapies which mostly destroy cancer cells [20]. The “single-hit kinetics” of phages, meaning only one phage is needed to target one cell, allows for fewer units of phages per treatment compared to their chemical counterparts.

Q2. The potential applications of phage display in cancer therapy are well described, particularly its role in targeted delivery and immune response activation. To reinforce the significance of this technology, consider adding a sentence on how phage display could overcome current challenges in cancer treatment.

--Ans: We revise the final sentence to create a stronger conclusion emphasizing the significance of phage display technique in phage treatment development: “Phage display can overcome lack of specificity in cancer treatment by leveraging antigen-receptor interaction and serving as a versatile vehicle for targeted delivery.”

  1. Main mechanisms of phage-based therapies

Q1. Consider briefly mentioning the potential advantages of using phages in this context, such as the ability to present a wide array of tumor-specific antigens or the potential for personalized cancer vaccines.

--Ans: To describe the significance of using phage to elicit an immune response, the sentence is added: “Phage display allows mass display of tumor-specific antigens with potential for vaccine development.”

Q2. The section on gene therapy is detailed and provides a good overview of different therapeutic strategies, including apoptosis induction, gene replacement, and prodrug activation. However, the text could be made more engaging by highlighting the potential impact of these strategies on patient outcomes, such as increased survival rates or reduced side effects. Consider discussing some of the challenges or limitations of these approaches, such as potential off-target effects or the need for precise delivery mechanisms, and how phages might help overcome them.

--Ans: Despite the theoretical advantages of gene therapy over conventional treatments, due to a lack of promising research, it is difficult to state that gene therapy has increased survival rates or reduced side effects.  However, we have mentioned the limitations of gene therapy and phage therapy can be used to overcome them: “Limitations of gene therapy include targeting inefficiency, vector instability, and infection risks from viral vectors; however, they can be overcome by using phages because of higher specificity through phage display, sound structure, and inability to infect mammalian cells.”

  1. Discussion

Q1. After discussing each challenge, it could be useful to briefly mention current strategies or research directions aimed at overcoming these issues.

--Ans: We have restructured this section by first addressing each challenge then mentioning strategies or research directions to overcome them: “While the growing interest in phage-based cancer therapies engenders hope, there are limitations that must be overcome for successful therapy. First are challenges that phage vectors face during circulation. The small nature of phages can trigger processing by the reticuloendothelial system (RES), which consists of phagocytes that phagocytose foreign particles [89]. Once the phages are cleared by the RES, they are destroyed by macrophages in the spleen or liver, decreasing their half-life. However, there are certain phage variants which can be used, termed long-circulating phages, that can circumvent the RES system and remain in blood circulation for longer (because of a mutation in the capsid) [7,17]. There may be impaired diffusion of vectors, especially into cores of solid tumors, due to high interstitial pressure or the unconventional features of angiogenic vessels; however, vascular-modulating drugs can be administered before phage treatment to overcome these issues [90]. Therapies against multidrug resistant tumors must also bypass the efflux systems that pump anticancer drugs from their cytosol to prevent treatment failure: further research should be conducted to determine vectors that can bypass this system [91]. To reduce challenges during the engineering process of phage constructs, antigen size and phage genome length should be kept reasonable to ensure viable phage particles. Lastly, mutations in receptors could render phage therapy ineffective by disrupting antibody-receptor interaction and losing selectivity. For instance, HER-2 mutations in breast cancer have led to resistance towards drugs such as trastuzumab due to alterations in HER-2 structure [92]. However, it is possible to engineer phages with antigens of known variants to overcome this challenge."

Q2. To reinforce the potential of phage-based therapies in targeting TAMs, you could compare the specificity and manipulability of phages with other TAM-targeting strategies, emphasizing the unique advantages phages might offer.

--Ans: We discuss the faults in current TAM-targeting targeted therapies and emphasize phages as a new potential nanosize vector worthy of consideration: “Although there have been several targeted therapies that have worked on animal models by blocking pathways and intracellular interactions, human patients often respond poorly [95], mostly due to lack of M2-specificity leading to toxicity for healthy cells [96]. With their selectivity and manipulability, phages could serve as efficient vectors for TAM-based treatment. In fact, Cieslewicz et al.’s study reported the discovery of a peptide M2pep that can preferentially target M2 macrophages with low impact on healthy leukocytes in murine models [97]. Although various nanoparticles are currently being developed to increase targeting specificity [95], phages should be taken into greater consideration for their easy and low cost of production."

Reviewer 3 Report

Comments and Suggestions for Authors

Dear authors

I make some suggestions for improve the manuscript.

ABSTRACT  SECTION 

Line 10. Please change “manip-ulability” by “manipulability”.

Line 16. Please change “thera-pies” by “therapies”.

Could be added a small explication What is the function of bacteriophages by infected specific human cells even cancer specifically, if the bacteriophages infect bacteria microorganisms only, may be used techniques of biology synthetic etc...please clearly explain it.

Introduction section

Line 47-48. Please change “chem- otherapy” by “chemotherapy”.

Line 48. Please change “se- quencing” by “sequencing”.

Section 2. What are bacteriophages?

Line 57. Please improve the description about taxonomical ICTV for bacteriophages including the class, families and highlight the families relevant for clinical treatment cancer.

Line 62-63.  please rewrite these sentences, because it is better to include the firsts experiments where the therapy was used by therapy anti cancer.  

Line 63-67. Please include an introduction description of current molecular techniques and strategies used  based on phages used as cancer therapy.

Line 73. Please check this sentence “Phages can have a narrow or broad host range”, because generally the phages are host specific, but could add an example of specific phage  and one of a phage broad used as cancer therapy.

Lines 76-81. Please check this section, because describe the general information about bacteriophages when infecting bacteria, it is better to add an explanation about how the basic knowledge about techniques of infection have been improved and today are a proposal for treatment anti-cancer, the phages that infected bacteria can infect cancer cells?  

and add a final sentence that is binding with the next section.

Phage-base therapy section

Line 106-140. Please check this section, I consider that  could be added examples that demonstrate the used phages by expressed proteins or antigens that activate an immune response specific to cancer cells.

Figure 1.Please here the figure caption should be improved, include the nomenclature description and include in the figure subsection (a,b,c, etc) and in the figure caption the description of these subsections.

Lines 152-156. Here explain the mechanism of phage based cancer therapy, could be included in the introduction and abstract because it is the relevant part of the manuscript, check it, please.

Line 158-205. Please here add the citation (Figure 1) with subsections added respectively.

Line 176. Here mentioned “recognize specific receptors”, Which?, could be explained more, including examples used currently for cancer therapy.

section 5.

Lines 220. Please add more details about examples described, for example include value of reduction about tumor reported by Wang et al (line 237).

Please check that in the next subsection about different cancers do not include the reference Table 1  in where corresponds.

Line 339.Please add what pages are associated here? (include names).

This section described a great number of examples and cancers which highlight the use of the phage in the therapy, however Observed that only the M13 and lambda phage are used in combination with other viruses used for building vaccines.

I suggest performing a table summarizing the information of different examples among cancer studies, it could include the following columns: kind  cancer, phage-based strategy, key molecule, cancer effect (include value prolificity or reduction measure tumors).

Discussion section.

Line 525. Please clearly describe that the use of treatment authorized in Russia and Georgia is for bacterial diseases and in the Uk, Belgium, Australia, India, China and US also used the therapy with bacteriophages in cases of ultra resistance bacteria disease. The phage therapy applied to cancer is  yet not authorized around the world. Please rewrite these final sections and make it. 

Please Include a conclusion section where discuss the difference between genic therapy vs phage-based therapy and highlight the potential application of vaccines design with and without phages.

Comments on the Quality of English Language

Moderate editing of English language required.

Author Response

Dear authors

I make some suggestions for improve the manuscript.

ABSTRACT 

Q1.Line 10. Please change “manip-ulability” by “manipulability”.

--Ans: We have changed “manip-ulability” to “manipulability

Q2.Line 16. Please change “thera-pies” by “therapies”.

--Ans: We have changed “thera-pies” to “therapies”.

Q3.Could be added a small explication What is the function of bacteriophages by infected specific human cells even cancer specifically, if the bacteriophages infect bacteria microorganisms only, may be used techniques of biology synthetic etc...please clearly explain it.

--Ans: We have added the description of the techniques used in phage- based cancer therapy in Abstract : “Common mechanisms are immunogenic antigen display on phage coats and the use of phage as transporters to carry drugs, genes, and other molecules”.

Introduction section

Q1.Line 47-48. Please change “chem- otherapy” by “chemotherapy”.

--Ans: We have changed “chem-otherapy” to “chemotherapy”.

Q2.Line 48. Please change “se- quencing” by “sequencing”.

--Ans: We have changed “se-quencing” to “sequencing”.

Section 2. What are bacteriophages?

Q1.Line 57. Please improve the description about taxonomical ICTV for bacteriophages including the class, families and highlight the families relevant for clinical treatment cancer.

--Ans: We have added a new Table.

Table 1. Classification and characterization of bacteriophages used in cancer therapy

Phage

type

Class

Family

Genus

Genome

Size

(kb)

Proteins

Morphology

Life cycle

(in bacteria)

M13

Faserviricetes

Inoviridae

Inovirus

ssDNA

6.4

11

Filamentous

Lysogenic

λ

Caudoviricetes

Siphoviridae

Lambdavirus

dsDNA

48.5

73

Icosahedral head with tail

Lytic/

lysogenic

T4

Caudoviricetes

Straboviridae

Tequatrovirus

dsDNA

168

289

Icosahedral head with tail

Lytic

T7

Caudoviricetes

Autographiviridae

Teseptimavirus

dsDNA

40

55

Icosahedral head with tail

Lytic

MS2

Leviviricetes

Fiersviridae

Emesvirus

ssRNA

3.6

4

Icosahedral

Lytic

Qβ

Leviviricetes

Fiersviridae

Qubevirus

ssRNA

4.2

4

Icosahedral

Lytic

Q2.Line 62-63.  please rewrite these sentences, because it is better to include the firsts experiments where the therapy was used by therapy anti cancer.  

--Ans: We have added the earliest report of phage interaction with tumor tissue:

“Although phages infect bacteria, they can also interact with eukaryotic cells. The  earliest report of phage interaction with tumor tissue was done in 1940 by Bloch et al. who observed that phages could accumulate in Erhlich carcinomas and inhibit tumor growth [15].”

Q3.Line 63-67. Please include an introduction description of current molecular techniques and strategies used based on phages used as cancer therapy.

--Ans: We have already included an introduction description of current molecular techniques and strategies used based on phage based cancer therapy in Section 3 : “Production of these antigen-specific phages is straightforward thanks to phage   display, a technique which has revolutionized the application of phages in              biotechnologies. In phage display, the genetic material which codes for a target        polypeptide is fused with genes that express coat proteins, thus allowing the polypeptide to be expressed on the surface of the phage [29-31]” ;  “Using this technique, scientists can create a large collection of phages (called a library) displaying different types of proteins. Due to the manipulability of phage coat protein expression, phage display libraries have been used to isolate target proteins, study protein-protein interactions, and identify bacterial strains [31].

Q4.Line 73. Please check this sentence “Phages can have a narrow or broad host range”, because generally the phages are host specific, but could add an example of specific phage and one of a phage broad used as cancer therapy.

--Ans: We have added examples in section 2: “Phages can have a narrow or broad host range (the diversity of organisms that the phage can infect). For example, JHP phage can infect P. aeruginosa, E. coli, S. enterica, Campylobacter jejuni, Acinetobacter baumanii and Proteus mirabilis, whereas M13 phage can only infect E. coli [9].”

Q5.Lines 76-81. Please check this section, because describe the general information about bacteriophages when infecting bacteria, it is better to add an explanation about how the basic knowledge about techniques of infection have been improved and today are a proposal for treatment anti-cancer, the phages that infected bacteria can infect cancer cells? 

and add a final sentence that is binding with the next section.

--Ans: We have added the basic knowledge about techniques of infection: “Research has led to a general understanding of phage-eukaryotic cell interactions which thus prompted the proposal of phage treatment for human cancer. Phages can enter eukaryotic cells through processes like receptor-mediated endocytosis, transcytosis, clathrin-mediated endocytosis, macropinocytosis, or caveolae-mediated endocytosis, but this depends on the type of eukaryotic cell and phage [16]. While phages do not infect and replicate within eukaryotic cells [17], they may still interact with intracellular proteins such as Toll-like receptors and cytosolic proteins [16].”

Phage-base therapy section

Q1.Line 106-140. Please check this section, I consider that could be added examples that demonstrate the used phages by expressed proteins or antigens that activate an immune response specific to cancer cells.

--Ans: We have added example which demonstrated the specific immune response induced by the display antigens on phage in Section 3: “An example polypeptide is the HER2 antigen which can induce anti-HER2 antibodies in breast cancer treatment.”

Q2.Figure 1.Please here the figure caption should be improved, include the nomenclature description and include in the figure subsection (a,b,c, etc) and in the figure caption the description of these subsections.

--Ans: We have added labels (component of phage, prodrug, TRAIL, antibody) and their descriptions in Figure 1. Figure 1. Mechanisms of phage-based cancer therapies (using filamentous phage as an example). (A) Cancer cell antigens can be displayed on phage surface using peptide display technique. These antigens can trigger an immune response, encourage production of anti-cancer antibodies, or activate cytotoxic T cells against cancer cells. (B) Phages can be used as transporters to deliver photosensitizers, cytokines or transgenes to cancer cells. APC, antigen presenting cell; TRAIL, tumor necrosis factor-related apoptosis-inducing ligand.”

Q3.Lines 152-156. Here explain the mechanism of phage based cancer therapy, could be included in the introduction and abstract because it is the relevant part of the manuscript, check it, please.

--Ans: We have added the mechanism of phage- based cancer therapy in the abstract: “Common mechanisms are immunogenic antigen display on phage coats and the use of phage as transporters to carry drugs, genes, and other molecules”.

Q4.Line 158-205. Please here add the citation (Figure 1) with subsections added respectively.

--Ans: We have added the citations of Figure 1 in Section 4 as mentioned : “The first mechanism is for the phage to elicit an immune response to destroy target cancer cells (Figure 1.A)”;  “The helper T cells pass along the antigen information to B cells, activating the B cells and encouraging production of antibodies that can bind to tumor cells with the same antigen on their cell surface (Figure 1.A.1)”;  “The cytotoxic T cells acknowledge the antigen and use this information to attack tumor cells displaying the same type of antigen through secretion of proteins such as granzyme, perforin, cathepsin C and granulysin to cause cell death [33] (Figure 1.A.2)”; “Phages can also be used as transport vehicles for targeted delivery (Figure 1.B)”; “Upon activation by light at a certain wavelength, these photosensitizers react with oxygen to form reactive oxygen species which can kill cancer cells (Figure 1.B.1)”; “Cytokines can also be displayed on phages to trigger immune responses including stimulation of macrophage inflammatory response [36] (Figure 1.B.2)”;  “But perhaps the most promising form of targeted delivery is gene therapy, which is the transfer of engineered genetic material into cells to alter cellular responses and treat diseases [37] (Figure 1.B.3)”;  ”Secreted TRAIL binds to  membrane receptors DR4 and DR5 which activates caspases or induces mitochondrial-dependent death [40] (Figure 1.B.3.a)”;  “ A faulty, mutated gene can also be replaced by inserting a  functional copy: this can be seen with gene therapies targeting p53 where a gene encoding functional p53 is delivered to cancer cells with mutated p53 (which has lost its tumor  suppressor functions) [41] (Figure 1.B.3.b)”;  “The enzyme can activate a prodrug into a cytotoxic agent through metabolization (Figure 1.B.3.c)”.

Q5.Line 176. Here mentioned “recognize specific receptors”, Which?, could be explained more, including examples used currently for cancer therapy.

--Ans: We have added an example of “recognize specific receptors” in Section 4: “Since phages can be engineered to display certain peptides which recognize specific receptors, such as the RDG4C ligand which binds to ⍺v3 and ⍺v5 integrins on cancer cells, they serve as good carrier candidates.”

section 5.

Q1.Lines 220. Please add more details about examples described, for example include value of reduction about tumor reported by Wang et al (line 237).

--Ans: We have added more details of Wang et al study in Subsection 5.1. “Within 15 weeks of age, all female mice treated with empty phages developed palpable tumors while 75% of mice vaccinated with ECTM-phages and 40% of mice vaccinated with Δ16ECTM-phages were still tumor-free.”

Q2.Please check that in the next subsection about different cancers do not include the reference Table 1  in where corresponds.

--Ans: The next subsection (which is 5.2 Colorectal cancer) has 3 references [51,38,52]. Refence 38 and 52 are shown in Table 2, but not Reference 51, which is general knowledge related. “Colorectal cancer (CRC), the second leading cause of cancer death in the world, is usually treated with surgical resection, chemotherapy, and radiation. Metastasis to organs like the liver and the lungs continues to be a challenge of CRC [51].”

Q3.Line 339.Please add what pages are associated here? (include names).

--Ans: We further clarified the association of “M13” phage as mentioned:

“ The current gold standard of idiotype vaccines is to chemically link idiotype to immunogenic protein keyhole limpet hemocyanin (KLH) [68], but recently there are studies where idiotype is being linked to M13 phage.”

Q4.This section described a great number of examples and cancers which highlight the use of the phage in the therapy, however Observed that only the M13 and lambda phage are used in combination with other viruses used for building vaccines.

I suggest performing a table summarizing the information of different examples among cancer studies, it could include the following columns: kind cancer, phage-based strategy, key molecule, cancer effect (include value prolificity or reduction measure tumors).

--Ans: We have added more types of phages studies below, and updated Table 2.

“5.1 Bresast cancer:

In 2008, Shadidi et al. showed that oral immunization with T7 phage displaying the tumor antigen Hsp27 could lead to the induction of effective immune responses in 4T1 murine breast cancer cells which overexpress Hsp27. Mice were subcutaneously injected with 4T1 cells on day 30 after immunization with the T7 phage expressing the Hsp27. Mice orally immunized with T7 phage expressing the Hsp27 developed tumors that had a significantly smaller average weight versus the wild-type phage (1.5±0.2g versus 2.5±0.5g; p<0.05). Additionally, mice immunized with the wild-type phage had significantly higher number of lung metastases than Hsp27 phage treatment (80±15 vs 40±10, p< 0.05). These data provided the first evidence that a recombinant T7 phage stably expressing Hsp27 can inhibit outgrowth of 4T1 breast cancer cells [49].

In Pouyanfard et al.’s 2012 study, multivalent T7 bacteriophage nanoparticles were developed that displayed an immunodominant H-2kd-restricted CTL epitope (9-mer  peptide p66 TYVPANASL) derived from the rat HER2/neu. The authors tested the therapeutic potential of these chimeric T7 nanoparticles in 6- to 8-week-old female BALB/c mice. Only T7 phage nanoparticles carrying a single copy of p66 peptide were successfully cross-presented and elicited a specific T cell response against the displayed CTL epitope. Immunization with these nanoparticles induced CTL-mediated lysis of P815 target cells in vitro and increased IL-4 cytokine secretion. T7-p66-vaccinated mice efficiently rejected HER2-expressing TUBO Cells. Moreover, anti-tumor effects and dramatic tumor regression were significant in some mice vaccinated with p66+FA, whereas all control mice developed large tumors by day eighty [50].

5.3 Lung adenocarcinoma

In Ren et al.’s study, a T4 phage nanoparticle expressing mouse Flt4 (mFlt4), which promotes tumor metastasis by stimulating solid tumor lymphangiogenesis, was constructed to evaluate the phage's antitumor activity. The effect of T4-mFlt4 vaccine was explored in mice injected with cells of Lewis lung carcinoma (LLC), a VEGFR-positive tumor. Mice with LLC-derived tumors showed extended survival when treated with the T4-mFlt4 vaccine compared to the control group. The treated group had a survival duration of 64 days, compared to 29 and 33 days in the control groups. By day 62, the survival rate in the T4-mFlt4 group was 25%, while the control phage group had a survival rate of 12.5% at 29 days. The vaccine did not demonstrate an ability to inhibit tumor growth, but it suppressed tumor metastasis (0.45 g±0.12 g versus 0.84 g±0.29 g in the   control, P=0.002) in mice. Histochemical examinations showed that the mean lymphatic microvessel counts were reduced in tumors after T4-mFlt4 treatment, but the average vascular microvessel counts in lungs were not significantly different between the groups [57].

Ren et al.’s 2011 study demonstrated the protective immunity of T4-mVEGFR2   vaccine against Lewis lung carcinoma (LLC) in mice. In vitro studies showed that immunoglobulin induced by T4-mVEGFR2 inhibited VEGF-mediated endothelial cell proliferation and tube formation. The antitumor activity was further confirmed through adoptive transfer of the purified immunoglobulin. The antitumor activity and production of autoantibodies against mVEGFR2 were neutralized when CD4+ T lymphocytes were depleted [58].

Zuo et al.’s 2019 study evaluated the anti-angiogenic effects of recombinant T4 phages expressing the extracellular domain of VEGFR2 (T4-VEGFR2). The T4-VEGFR2 phages specifically bound to VEGF and inhibited VEGF-induced phosphorylation of VEGFR2 and downstream kinases, such as extracellular signal-regulated kinase (ERK) and p38 mitogen-activated protein kinase (MAPK). In vitro experiments demonstrated that T4-VEGFR2 phages could inhibit VEGF-induced endothelial cell proliferation and migration. The administration of T4-VEGFR2 phages suppressed tumor growth, reduced microvascular density, and prolonged survival in murine models of LLC and colon carcinoma (CT26 cell line) [59].

5.4 Hepatocellular carcinoma

Chang et al.’s 2016 study developed a delivery system using MS2 virus-like particles (VLPs) crosslinked with the GE11 polypeptide. MS VLPs delivered long non-coding RNA maternally expressed gene 3 (MEG3) specifically to epidermal growth factor receptor (EGFR)-positive hepatocellular carcinoma (HCC) cell lines, without activating EGFR downstream signaling pathways. It significantly inhibited tumor cell growth (HepG2, Hep3B, Huh7) and invasion (HepG2) both in vitro and in mice, indicating that MS2 VLP is a promising approach for long non-coding RNA -based cancer therapy [64].

In 2021, Zhang et al. explored the anti-tumor effects of two microRNAs using MS2 VLPs containing the miR-21 sponge and pre-miR-122 sequences. These VLPs were crosslinked with a peptide targeting HCC cells. The VLPs delivered miR-21 sponge into cells, and the linked pre-miR-122 was processed into mature miR-122. The miR-21 sponge inhibited the proliferation, migration, and invasion of HCC cells. Simultaneous delivery of miR-21 sponge and miR-122 further suppressed proliferation (34%), migration (63%), and invasion (65%), while promoting apoptosis in HCC cells. These results indicate that MS2 VLPs-delivered microRNAs are also efficient therapeutic approaches for HCC [65].

5.9 Melanoma

Shukla et al. in 2018 reported the preclinical study of immunization with T7 phage nanoparticles that display murine melanoma B16F10 neoantigens to elicit plasma antibody and vaccine-draining lymph node B cell responses. They found that a single   injection of T7 peptide vaccine was able to stimulate an anti-peptide response, and shorter peptides with 11 amino acids stimulated antibodies to the mutated epitope better than longer peptides with 34 amino acids. A trimer of 11 repeated amino acids provided no advantage over single monomer as a T7 expressed vaccine [79].

Hwang and Myung et al.’s study in 2020 tested T7 phage displaying a peptide able to target B16F10 cells. This engineered T7 phage contained a mammalian expression cassette of the cytokine granulocyte macrophage-colony stimulating factor (GM-CSF). GM-CSF was expressed in the transduced cells in vitro and in vivo. Mice treated with the phage intravenously survived for 25 days, whereas only 40% of untreated mice survived. Over the 16 days of treatment, this engineered T7 phage reduced tumor growth by 72% compared to untreated controls. Serum levels of IL-1α, TNF-α, and GM-CSF increased during treatment. By altering the microenvironment and promoting the recruitment of anti-tumor immune cells (macrophages, dendritic cells (DCs), and CD8+ T cells), recombinant T7 phage effectively inhibited tumor growth [80].

In 2023, Rashidijahanabad et al. reported a novel anticancer vaccine by covalently conjugating sialyl Lewisa, an attractive carbohydrate-associated cancer antigen to bacteriophage Qβ. The sera from SLewisa-Qβ immunized mice exhibited higher anti-SLewisa antibody titers, which remained detectable at high levels even after 379 days. C57BL/6 mice were immunized with Qβ-sLea conjugate or control (Qβ+sLea admixture), and injected with B16 melanoma cells stably expressing fucosyltransferase 3 (B16FUT3, which led to cell surface production of sLea). The mice vaccinated with Qβ-sLea conjugate had significantly lower numbers of tumor foci in lungs compared to the group         administered with the control [81].

Table 2.

phage

Peptide display

Mechanism

Result

Year

Breast cancer

(BALB/c mice)

T7 phage

Hsp 27

Induction of antibodies

Significantly developed smaller average tumor weight (1.5± 0.2g vs 2.5± 0.5g) in mice, which were

subcutaneously injected with 4T1 breast carcinoma cell.

2008 [49]

Lung carcinoma

(mice)

T4 phage

mFlt4

Induction of antibodies

Prolonged survival in mice, which were injected with Lewis lung

carcinoma cells

2009 [57]

Lung carcinoma

(C57BL/6J mice)

T4 phage

mVEGFR2

Induction of antibodies

Obvious inhibition of tumor growth and significant extension of survival length

2011 [58]

Breast cancer

(BALB/c mice)

T7 phage

H-2k(d)-restricted CTL epitode

(derived from rat HER2/neu)

Induction of cytotoxic

immune response

Protected mice against HER2

positive tumor challenge in both prophylactic and therapeutic

setting

2012 [50]

Hepatocellular

Carcinoma

(BALB/c mice)

MS2 phage

GE11 peptide

Introducing IncRNA MEG3  transgene into EGFR positive HCC cell lines

Significant inhibition of

tumor growth

2016 [64]

Lung carcinoma

(BALB/c or C57BL/6 mice)

T4 phage

VEGFR2

VEGFR2 delivery (binding to VEGF), altering tumor microenvironment

Suppress tumor growth and decrease microvascular density in murine models of Lewis lung carcinoma

2018 [59]

Melanoma

(mice)

T7 phage

11-AA and 34-AA peptides

Induction of antibodies

11-AA peptides showed better  antibody stimulation than longer peptides (34-AA peptides)

2018 [79]

Melanoma

(BALB/c mice)

T7 phage

pep42

Introducing expression cassette of GM-CSF transgene into cancer cells

Inhibited tumor growth (B16F10 melanoma cells) by 72% compared to untreated control

2020 [80]

Hepatocellular

Carcinoma

(HCC cell line)

MS2 phage

GE-11 peptide

Introducing microRNA-21-sponge and pre-microRNA-122 transgenes into HCC cells

Decreased proliferation, migration and invasion of HCC cells

2021 [65]

Melanoma

(mice)

Qβ phage

Sialyl Lewis antigen

Induction of antibodies

Significantly reduced tumor development in mice metastatic cancer model

2023 [81]

Discussion section.

Q1.Line 525. Please clearly describe that the use of treatment authorized in Russia and Georgia is for bacterial diseases and in the Uk, Belgium, Australia, India, China and US also used the therapy with bacteriophages in cases of ultra resistance bacteria disease. The phage therapy applied to cancer is yet not authorized around the world. Please rewrite these final sections and make it. 

--Ans: Corrections have been made as follows:

Several countries are at varying stages of integrating phage therapy into their medical treatments offered. Except for Russia and Georgia, where phage cocktails such as “Pyophage” and “Intestiphage” to treat bacterial infection are available for consumer  purchase without prescription [98], most people access phage therapy through       compassionate use, which is when seriously ill patients can be treated with an          investigational product if there are no clinical trials or safer options available [99].   Countries such as the UK, France, Belgium, Australia, India, China, and the US allow the use of compassionate treatment for severe resistant infectious disease. Another         administration of phage-based treatment is through magistral preparation, a personalized approach where the final product is compounded in a pharmacy based on a prescription: this can be seen in Poland, Georgia, and Belgium [100,101].  Although there is no FDA   approved phage therapy in the US thus far, the US is making significant strides in phage research, with the most phage-related investigator-sponsored trials (IST) compared to other countries (many of these trials have advanced to phase 3 status) [101]. While countries like India are without clear regulations, most countries are beginning to acknowledge the potential of phage therapy and are pushing for official legislation or research. It is evident that the world is starting to embrace the prospects of phage therapy, although its application to cancer treatment has still not been authorized in any country.

Q2.Please Include a conclusion section where discuss the difference between genic therapy vs phage-based therapy and highlight the potential application of vaccines design with and without phages.

--Ans: We have summarized our thoughts of your suggestions in the discussion section: “To date, gene and vaccine therapy have gained great interest in scientific research for cancer therapy which could be well explored using different vectors. Compared with other vectors, phages stand out for their high specificity, reduced toxicity, and low cost of production. Ongoing in-depth research of phage-based therapy is no doubt worthwhile and marks an exciting turn in the biomedical sciences as we discover novel applications of phage to treat cancer and other health conditions.”

Round 2

Reviewer 3 Report

Comments and Suggestions for Authors

Dear authors

This version manuscript is more clear and considerate that all comments were attended. The table 1 complement perfectly the context explained in the manuscript.

In my opinion  for this version is complete an is excelent, congratulations.